# Torus Graphs for Large Scale Neural Phase Analysis

**Jack Goffinet** [1]   **Casey Hanks** [1]   **David E. Carlson** [1][2]

## Abstract

Oscillatory neural signals such as electroencephalography (EEG) and local field potentials (LFPs) show phase relationships that coordinate communication across brain regions. Modern recordings capture hundreds of channels across many frequency bins, yet standard phase analyses are restricted to only a few variables. The Torus Graph (TG) model, an exponential-family distribution over phases whose univariate and pairwise potentials generalize von Mises distributions, infers principled structure among oscillations but models only static, undirected dependencies and is limited to $\sim 100$ variables because its score matching inference scales as $\mathcal{O}(d^6)$. We introduce a stochastic score matching procedure that reduces the per-iteration cost to $\mathcal{O}(d^2)$, enabling inference on datasets with thousands of variables. This scalable foundation supports analyses of 1,860 frequency-phase features from multielectrode LFPs and enables two extensions previously inaccessible to TGs or classical circular statistics: (i) a TG Hidden Markov Model capturing state-dependent phase-coupling changes (e.g., spindle-related states during sleep) and (ii) an autoregressive TG inferring directional interactions via transfer-entropy estimation. Applied to LFP recordings, these models reveal state-dependent phase-interaction patterns between wakefulness and NREM sleep. Together, they enable systematic, large-scale mapping of dynamic and directional phase relationships across brain and cognitive states.

[1]Department of Computer Science, Duke University, Durham NC, USA [2]Departments of Civil and Environmental Engineering; Biostatistics and Bioinformatics; and Electrical and Computer Engineering, Duke University, Durham NC, USA. Correspondence to: Jack Goffinet <jack.goffinet@duke.edu>.

*Proceedings of the 43$^{rd}$ International Conference on Machine Learning*, Seoul, South Korea. PMLR 306, 2026. Copyright 2026 by the author(s).

## 1. Introduction

Electroencephalography (EEG) and local field potential (LFP) signals are commonly described as the superposition of oscillations, each characterized by a phase that advances at a near-constant rate. Phase has long been recognized as a core computational variable in neural systems, leading to discoveries such as hippocampal theta phase precession in place coding (O'Keefe & Recce, 1993), phase–amplitude coupling in the entorhinal cortex (Chrobak & Buzsáki, 1998), the theory of communication-through-coherence in the visual cortex (Fries et al., 2001), and the dependence of visual perception on the prestimulus phase of ongoing scalp EEG oscillations (Busch et al., 2009). Such findings have motivated broader computational investigations into the role of neural phase in cognition (Buzsaki & Draguhn, 2004; Wang, 2010).

More recently, advances in large-scale neural recording technologies now enable the collection of high-dimensional electrophysiology data from hundreds of channels, each with up to hundreds of frequency bands (Stringer et al., 2019; Steinmetz et al., 2021). These developments have enabled opportunities to study complex, large-scale phase relationships across brain regions and frequency bands. Influential models of neural synchronization, most prominently Kuramoto oscillator systems (Kuramoto, 1984; Breakspear et al., 2010) and related frameworks (Winfree, 1980; Ermentrout & Kleinfeld, 2001), have been used extensively to study large-scale coordination, metastability, and collective dynamics in the brain (Bick et al., 2020). However, empirical analyses of LFP recordings still predominantly rely on amplitude-based measures or pairwise phase metrics such as Phase Locking Value (PLV) (Lachaux et al., 1999) or coherence (Srinath & Ray, 2014). These pairwise metrics and increasingly large datasets also introduce significant analytical challenges: the dimensionality of modern datasets far exceeds the capacity of traditional circular statistics or pairwise connectivity measures, leaving high-dimensional and multivariate phase-only structure relatively unexplored.

Despite growing interest in phase data across neuroscience and related fields, practical and flexible statistical methods for analyzing such data remain underdeveloped. Classical circular statistics do not scale beyond a few variables; for example, the PLV, widely used in neuroscience, is restricted to

pairwise phase comparisons. Pairwise metrics such as PLV are also statistically limited in a more fundamental sense: they cannot distinguish direct from indirect interactions. For example, Klein et al. (2020) show that PLV indicates synchrony between dentate gyrus and prefrontal cortex in rodent LFP recordings, yet these regions are conditionally independent given subiculum, which acts as an intermediary. Without accounting for such indirect pathways, pairwise connectivity graphs will systematically contain spurious edges, complicating neuroscientific interpretation. More recently, the Torus Graph (TG) distribution was introduced as a principled exponential family model on the torus, with univariate and bivariate potentials analogous to those in the multivariate normal distribution (Klein et al., 2020). TGs enable conditional independence estimation in multivariate phase data, an ability unavailable to pairwise measures such as PLV. However, fitting TGs is computationally demanding: inference scales poorly with dimensionality, effectively limiting applications to fewer than 100 variables in practice. While discriminative and self-supervised approaches have made significant progress on large-scale EEG classification and BCI tasks (Tang et al., 2022; Mohammadi Foumani et al., 2024; Kotoge et al., 2025), our goal is distinct: rather than optimizing performance on a downstream prediction task, we seek a generative probabilistic model over circular phase variables that supports conditional independence inference, dynamic coupling, and directional interaction estimation.

Beyond the statistical motivation above, there is also theoretical and practical justification for focusing exclusively on phase. Phase reduction theory (Winfree, 1980) shows that when oscillatory activity is well-approximated by stable limit cycles, amplitude dynamics rapidly contract, making phase the primary degree of freedom governing long-term dynamics. Thus, under certain assumptions, a phase-only representation captures the relevant dynamical structure. In practice, focusing on phase also helps avoid amplitude-related confounds such as electrode referencing effects and frequency-dependent gain in recording hardware, both of which complicate the interpretation of power-based measures (Srinath & Ray, 2014).

Our motivation is therefore to investigate what insights become accessible when analysis is conducted entirely in the phase domain, using probabilistic models capable of capturing conditional independence structure, dynamic coupling, and directional interactions in high-dimensional toroidal data.

In this work, we extend Torus Graphs to be **efficient**, **dynamic**, and **directional**. First, we introduce a stochastic score matching approach that enables scalable inference on datasets with thousands of phase variables. Building on this foundation, we develop a hidden Markov model variant

(TG-HMM) to capture time-varying connectivity among oscillatory signals. Finally, we propose an autoregressive variant (AR-TG) that supports inference of directed dependencies and estimation of transfer entropy in multivariate phase systems. We evaluate these models on both synthetic data and multi-electrode mouse LFP recordings, where they reveal dynamic oscillatory motifs and directional influences across brain regions. These contributions provide a general framework for scalable, probabilistic modeling of circular-valued data.

## 2. Background

### 2.1. Phase Locking Value

The most common phase coupling measure is the phase locking value (PLV), defined between pairs of random phase variables $X$ and $Y$ (Lachaux et al., 1999):

$$PLV_{X,Y} = |\mathbb{E}\, e^{i(X-Y)}| . \tag{1}$$

If $X$ and $Y$ have consistent phase differences, their PLV attains a maximum value of 1, whereas inconsistent phase differences result in a minimum PLV of 0.

### 2.2. The von Mises Distribution

Define the $d$-torus by $\mathbb{T}^d = [0, 2\pi)^d$ and let $x \in \mathbb{T}^1$ be an arbitrary point on the circle (the 1-torus). The von Mises distribution is a fundamental distribution on the circle, defined by

$$p(x; \kappa, \mu) = \frac{\exp[\kappa \cos(x-\mu)]}{2\pi \mathcal{I}_0(\kappa)} , \tag{2}$$

where $\kappa \geq 0$ is a concentration parameter, $\mu \in [0, 2\pi)$ is a location parameter, and $\mathcal{I}_n$ is the modified Bessel function of the first kind at order $n$. The distribution is unimodal with mode $\mu$ except when it reduces to the uniform distribution at $\kappa = 0$. The mean value of $e^{iX}$ when $X \sim vM(\mu, \kappa)$ is given by $e^{i\mu}\, \mathcal{I}_1(\kappa)/\mathcal{I}_0(\kappa)$.

### 2.3. The Torus Graph Distribution

The Torus Graph (TG) is the maximum entropy distribution of phase variables subject to *i)* the first moment of each phase variable ($\mathbb{E}\cos x_j$, $\mathbb{E}\sin x_j$) and *ii)* second moments between distinct phase variables ($\mathbb{E}[\cos x_j \cos x_k, \cos x_j \sin x_k, \sin x_j \cos x_k, \sin x_j \sin x_k]$ for $j \neq k$) (Klein et al., 2020). We can rewrite the products of univariate trigonometric functions in terms of more interpretable phase sum and phase difference statistics. Let $\mathbf{x} \in \mathbb{T}^d$. The general form of the TG model is:

$$p(\mathbf{x}; \boldsymbol{\phi}) \propto$$

$$\exp\left[\sum_{j=1}^d \phi_j^\top \begin{bmatrix} \cos x_j \\ \sin x_j \end{bmatrix} + \sum_{j<k} \phi_{jk}^\top \begin{bmatrix} \cos(x_j - x_k) \\ \sin(x_j - x_k) \\ \cos(x_j + x_k) \\ \sin(x_j + x_k) \end{bmatrix}\right] \tag{3}$$

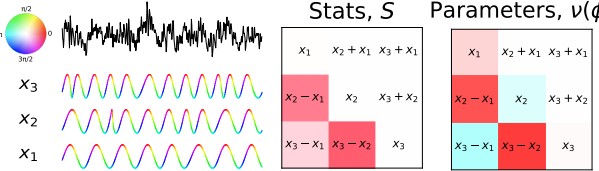

*Figure 1.* Visualization of TG statistics and parameters. **Left:** phases $x_1, x_2, x_3$ are extracted from a signal (black) via a continuous wavelet transform, with color denoting phase angle according to the colorwheel shown. **Right:** TG statistics and parameters are arranged as a $d \times d$ complex matrix, where each entry's cosine and sine components are treated as real and imaginary parts respectively, and colored by position in the complex unit disc. Diagonal entries correspond to univariate (single-phase) terms, lower-triangular entries correspond to phase-difference terms, and upper-triangular entries correspond to phase-sum terms. Parameters $\phi$ are passed through the normalization $\nu(z) = e^{i\angle z}\mathcal{I}_1(|z|)/\mathcal{I}_0(|z|)$ to map them to the unit disc before coloring. A white entry indicates a near-zero value.

Given $d$ phase variables, the model has $2d^2$ parameters. Define the following moment vector: $S(\mathbf{x}) = [S^1(\mathbf{x}), S^2(\mathbf{x})]^\top$ where $S^1(\mathbf{x}) = [\cos x_1, \sin x_1, \ldots, \cos x_d, \sin x_d]^\top \in \mathbb{R}^{2d}$ contains the univariate moments and $S^2(\mathbf{x}) = [\cos(x_1 - x_2), \sin(x_1 - x_2), \cos(x_1 + x_2), \ldots, \sin(x_{d-1} + x_d)]^\top \in \mathbb{R}^{2d(d-1)}$ contains the bivariate moments. We can then write the TG density more compactly as $p(\mathbf{x}; \phi) \propto \exp(\phi^\top S(\mathbf{x}))$.

The Torus Graph has three notable properties that we will make use of in this work. First, the TG is a multivariate generalization of the von Mises distribution. Second, the TG is closed under conditioning, so in particular the univariate conditionals are von Mises, enabling efficient Gibbs sampling from standard von Mises samplers. See Appendix A for conditioning formulas. Note, however, that the TG is not closed under marginalization, meaning $\mathbf{x} = [x_1; x_2]^\top \sim TG(\,\cdot\,; \phi)$ does not imply $x_1 \sim TG(\,\cdot\,; \phi')$ for some $\phi'$, an important difference from multivariate normal distributions. Lastly, the TG has an intractable normalizing constant. This prohibits practical maximum likelihood parameter estimation.

Several subfamilies of the TG have been introduced for specialized purposes, including those introduced by Zemel et al. (1992); Cadieu & Koepsell (2010); Mardia et al. (2007), which restrict the sufficient statistics to capture specific phase relationships, such as only phase difference terms. See Klein et al. (2020) for a more detailed discussion.

**Visualization** In this paper, we extract angles from one-dimensional signals such as LFPs by computing a continuous wavelet transform of the signal at different center frequencies, which decomposes the signal into oscillatory components and associates each with a complex-valued coefficient, and then taking the complex angle (i.e., $\angle z \in [0, 2\pi)$)

of each coefficient to obtain an instantaneous phase at each frequency (Figure 1, left). We visualize the TG statistics $\mathbb{E}[S(\mathbf{x})]$ by interpreting the cosine and sine statistics as real and imaginary parts, respectively, of a complex statistic arranged in a $d$-by-$d$ matrix. The diagonal entries of this matrix contain the univariate terms, the lower triangular entries contain the angle difference terms, and the upper triangular entries contain the angle sum terms. Each entry is then colored by its position in the complex unit disc given by the colorwheel in Figure 1. The TG parameters $\phi$ are visualized similarly, but by first applying the normalization $\nu(z) = e^{i\angle z}\mathcal{I}_1(|z|)/\mathcal{I}_0(|z|)$ elementwise to map terms to the complex unit disc (Figure 1, right).

**Score Matching for TGs** Due to the intractable normalizing constant of TG models, Klein et al. (Klein et al., 2020) propose fitting the parameters to data using a score matching approach (Hyvärinen, 2005). The score matching objective is

$$J(\phi) = \frac{1}{2} \int_{\mathbb{T}^d} p(x) \left\| \nabla_\mathbf{x} \log q(\mathbf{x}; \phi) - \nabla_\mathbf{x} \log p(\mathbf{x}) \right\|_2^2 \, d\mathbf{x} \,, \tag{4}$$

where $p(\mathbf{x})$ denotes the unknown data density and $q(\mathbf{x}; \phi)$ denotes the TG density with natural parameters $\phi$. Under mild regularity assumptions, Klein et al. show that this objective can be re-written as

$$J(\phi) = \mathbb{E}_{\mathbf{x} \sim p(\mathbf{x})} \left[ \frac{1}{2} \phi^\top \Gamma(\mathbf{x}) \phi - \phi^\top \mathbf{h}(\mathbf{x}) \right] \tag{5}$$

with $\mathbf{h}(\mathbf{x}) \triangleq [S^1(\mathbf{x}), 2S^2(\mathbf{x})]^\top \in \mathbb{R}^{2d^2}$ a vector of certain moments and $\Gamma(\mathbf{x}) \triangleq \nabla_\mathbf{x} S(\mathbf{x}) (\nabla_\mathbf{x} S(\mathbf{x}))^\top \in \mathbb{R}^{2d^2 \times 2d^2}$ the large outer product of the sufficient statistics Jacobian.

Setting the derivative of $J$ with respect to $\phi$ to 0 and solving for $\phi$ yields a unique closed-form solution when $\Gamma$ is invertible:

$$\frac{d}{d\phi} J(\phi) = \Gamma\phi - \mathbf{h} = 0 \;\Rightarrow\; \phi = \Gamma^{-1}\mathbf{h} \tag{6}$$

The solution is unbiased when $\Gamma = \mathbb{E}_\mathbf{x}[\Gamma(\mathbf{x})]$ and $\mathbf{h} = \mathbb{E}_\mathbf{x}[\mathbf{h}(\mathbf{x})]$ are estimated through samples. Then $\phi$ can be found through a simple linear solve after estimating $\Gamma$ and $\mathbf{h}$. However, $\Gamma \in \mathbb{R}^{2d^2 \times 2d^2}$, so the complexity of the linear solve operation via, e.g., Gaussian elimination scales extremely poorly as $\mathcal{O}(d^6)$.

## 3. Large Scale TGs

In this section we extend the TG methods proposed by Klein et al. to handle much larger datasets with many more variables. We additionally introduce conditional TGs, TG hidden Markov models, and transfer entropy estimation procedures for auto-regressive TG models.

### 3.1. Stochastic Score Matching for TGs

To avoid the $\mathcal{O}(d^6)$ scaling of fitting TG models through a linear solve, we resort to stochastic optimization of the score matching objective. Note that we can rewrite the $\Gamma$ term in Eq. 5 as $\phi^\top \Gamma(\mathbf{x})\phi = \phi^\top \nabla_{\mathbf{x}} S(\mathbf{x}) (\nabla_{\mathbf{x}} S(\mathbf{x}))^\top \phi = \|\phi^\top \nabla_{\mathbf{x}} S(\mathbf{x})\|_2^2$. Then we can rewrite the score matching objective (Eq. 5) as:

$$J(\phi) = \mathbb{E}_{\mathbf{x}} \left[ \tfrac{1}{2} \|\phi^\top \nabla_{\mathbf{x}} S(\mathbf{x})\|_2^2 - \phi^\top \mathbf{h}(\mathbf{x}) \right] \quad (7)$$

Although a dense Jacobian of $S(\mathbf{x})$ would have $\mathcal{O}(d^3)$ entries, each statistic depends on at most two phase variables, so the number of nonzero entries is only $\Theta(d^2)$. Accordingly, the term $\phi^\top \nabla_{\mathbf{x}} S(\mathbf{x})$ can be computed in $\mathcal{O}(d^2)$ time using a vector-Jacobian product (VJP), i.e. by applying reverse-mode autodiff to $\phi^T S(\mathbf{x})$ without ever forming the Jacobian. This operation dominates the per-iteration cost of optimization, resulting in an overall time complexity of $\mathcal{O}(d^2)$ per iteration. Of course, this time complexity is not directly comparable to the $\mathcal{O}(d^6)$ closed-form solve required by exact score matching (Eq. 6), since stochastic optimization is an iterative procedure. However, in practice stochastic score matching is far more scalable. Moreover, exact score matching requires storing $\Gamma \in \mathbb{R}^{2d^2 \times 2d^2}$, an $\mathcal{O}(d^4)$ object, whereas stochastic score matching requires only $\mathcal{O}(d^2)$ memory.

We implement stochastic score matching by sampling random minibatches, approximating the expectation in Eq. 7 with the batch average, and updating the parameters $\phi$ using the Adam optimizer (Kingma & Ba, 2015). The objective is compatible with both $L_2$ and group-$\ell_1$ regularization on $\phi$, the latter of which encourages sparse graph structure as discussed by Klein et al. (2020).

### 3.2. Stochastic Score Matching for Conditional TGs

Suppose that in addition to our phase variables $\mathbf{x} \in \mathbb{T}^d$, we observe covariates $y \in \mathbb{R}^m$. We wish to fit conditional models of the form $p(\mathbf{x}|y) \propto \exp(\phi(y)^\top S(\mathbf{x}))$. Let $\phi : \mathbb{R}^m \to \mathbb{R}^{2d^2}$ denote a possibly trainable map (e.g. a deep neural network) from covariates to TG natural parameters. In this case, the score matching objective is

$$J(\phi) = \mathbb{E}_{\mathbf{x},y} \left[ \frac{1}{2} \phi(y)^\top \Gamma(\mathbf{x})\phi(y) - \phi(y)^\top \mathbf{h}(\mathbf{x}) \right] \quad (8)$$

Our conditional stochastic score matching objective is a simple extension of the unconditional stochastic score matching objective (Eq. 7):

$$J(\phi) = \mathbb{E}_{\mathbf{x},y} \left[ \frac{1}{2} \|\phi(y)^\top \nabla_{\mathbf{x}} S(\mathbf{x})\|_2^2 - \phi(y)^\top \mathbf{h}(\mathbf{x}) \right] . \quad (9)$$

As in the unconditional case above, this approach provides an unbiased gradient estimator that can be computed in $\mathcal{O}(d^2)$ time per iteration.

### 3.3. Torus Graph Hidden Markov Model

We can use the stochastic score matching approach described above to infer the natural parameters $\phi$ of a large TG model, uncovering the conditional independence structure of the underlying process. However, it is widely believed that oscillations in the brain *dynamically* route information through the brain (Buzsáki, 2006), motivating a dynamic torus graph model. In this section we present a method to approximately fit a Torus Graph Hidden Markov Model (TG-HMM) with the ability to infer a discrete number of hidden states, each corresponding to a separate TG model, and a description of the Markov transition structure between them. Also note that the TG-HMM generalizes a mixture model of torus graphs, which can be fit by simply fixing an i.i.d. Markov transition structure.

The central difficulty in fitting a TG-HMM is the intractability of the normalizing constants. To use the Baum-Welch (forward-backward) algorithm as part of an Expectation Maximization (EM) procedure, we need the relative probability of every observation under different discrete states, which typically requires calculating intractable normalizing constants. We take an approach inspired by the noise-contrastive estimation literature (Gutmann & Hyvärinen, 2010), where we learn free parameters that approximate the log normalizing constants and use a logistic regression classifier to estimate the soft state assignments, facilitating tractable EM steps.

To explain the strategy, let $x_t \in \mathbb{T}^d$ for $t = 1, \ldots, T$ be our observations and let $z_t \in \{1, \ldots, K\}$ denote our hidden states with transition matrix $p(z_{t+1} = k | z_t = j) = \Pi_{jk}$. The emission for state $k$ is given by $p(x_t | z_t = k) = \exp(\phi_k^\top S(x_t) - A(\phi_k))$ where $A(\phi_k)$ is the unknown log normalizing constant of the $k^{\text{th}}$ TG. The joint log likelihood, omitting the initial state prior $p(z_1)$, which we assume to be uniform, is

$$\log p(z, x) = \sum_{t=2}^{T} \log \Pi_{z_{t-1}, z_t} + \sum_{t=1}^{T} [\phi_{z_t}^\top S(x_t) - A(\phi_{z_t})] . \quad (10)$$

We never compute the log normalizing constants $A(\phi_k)$, but instead introduce the free parameters $A_k \in \mathbb{R}$ for $k = 1, \ldots, K$, with light ridge regularization to keep them bounded and identifiable. These parameters then define an inexact surrogate joint model:

$$\log \tilde{p}(z, x) = \sum_{t=2}^{T} \log \Pi_{z_{t-1}, z_t} + \sum_{t=1}^{T} [\phi_{z_t}^\top S(x_t) - A_{z_t}] . \quad (11)$$

**E-Step** Compute the HMM posteriors

$$\gamma_{t,k} = \tilde{p}(z_t = k \mid x_{1:T}), \quad \xi_{t,i,j} = \tilde{p}(z_{t-1} = i, z_t = j \mid x_{1:T}) , \quad (12)$$

using the standard forward-backward procedure with the inexact surrogate model, not the true TG models with unknown normalizing constants.

**M-Step** Maximize the expected complete data log likelihood under these posteriors. First set

$$\Pi_{ij} = \frac{\sum_{t=2}^{T} \xi_{t,i,j}}{\sum_{j'=1}^{K} \sum_{t=2}^{T} \xi_{t,i,j'}} \ . \qquad (13)$$

Then for the emission update, we would like to maximize:

$$Q(\boldsymbol{\phi}) = \sum_{t,k} \gamma_{t,k}[\boldsymbol{\phi}_k^\top S(x_t) - A(\boldsymbol{\phi}_k)] \qquad (14)$$

We find a maximum of $Q$ by setting the derivative of $Q$ w.r.t $\boldsymbol{\phi}_k$ to 0 and solving:

$$\frac{\partial Q}{\partial \boldsymbol{\phi}_k} = 0 \quad \Rightarrow \quad \nabla A(\boldsymbol{\phi}_k) = \frac{\sum_t \gamma_{t,k} S(x_t)}{\sum_t \gamma_{t,k}} \qquad (15)$$

We now introduce a modified objective with the free parameters $A_1, \ldots, A_K$ taking the place of the unknown log partition function $A(\boldsymbol{\phi})$:

$$Q'(A) = \sum_{t,k} \gamma_{t,k}[\boldsymbol{\phi}_k^\top S(x_t) - A_k]$$
$$- \sum_t \log \sum_k \exp(\boldsymbol{\phi}_k^\top S(x_t) - A_k) \, . \quad (16)$$

We note that this objective corresponds to the multinomial logistic regression log likelihood with soft targets $\gamma_{t,k}$, features $S(x_t)$, fixed weights $\{\boldsymbol{\phi}_k\}$, and trainable class intercepts $-A_k$, which has a convex negative log likelihood. As above, we find the stationary point of $Q'$ by setting its derivative w.r.t $A_k$ to 0. We find

$$\sum_t \gamma_{t,k} = \sum_t \frac{\exp(\boldsymbol{\phi}_k^\top S(x_t) - A_k)}{\sum_j \exp(\boldsymbol{\phi}_j^\top S(x_t) - A_j)} \quad \text{for all } k. \quad (17)$$

Note that the softmax term need only match the $\gamma_{t,k}$ *in aggregate*, summed over timepoints $t$, and will not in general match for any specific time $t$.

Now let $r_{t,k} = p(z_t = k|x_{1:T})$ be the true HMM posteriors under the exact TG models and let $s_{t,k} = \text{softmax}_k(\{\boldsymbol{\phi}_j^\top S(x_t) - A(\boldsymbol{\phi}_j)\}_j)$ be the *emission-only* state probabilities under the true TG models. We will also consider $\tilde{s}_{t,k} = \text{softmax}_k(\{\boldsymbol{\phi}_j^\top S(x_t) - A_j\}_j)$, the emission-only state probabilities under the surrogate emission model and $\gamma_{t,k}$, the smoothed posteriors under the surrogate emission model, output by forward-backward. Denote the weighted empirical moments for any weights $w$ by $\hat{\mu}_k(w) = \sum_t w_{t,k} S(x_t) / \sum_t w_{t,k}$.

We will proceed to show that optimizing the modified objective $Q'$ approximately satisfies the stationary condition of the exact objective $Q$ (Eq. 15). First we have

$$\nabla A(\boldsymbol{\phi}_k) = \mathbb{E}_{\mathbf{x} \sim p(\mathbf{x}; \boldsymbol{\phi}_k)}[S(\mathbf{x})] \, , \qquad (18)$$

a general property of exponential families. Then, if the TG family is correctly specified

$$\mathbb{E}_{\mathbf{x} \sim p(\mathbf{x}; \boldsymbol{\phi}_k)}[S(\mathbf{x})] \approx \mathbb{E}_{\mathbf{x} \sim p_{\text{data}}(\mathbf{x}|z=k)}[S(\mathbf{x})] \, . \qquad (19)$$

Next, if we have good finite sample estimates of moments under the true posteriors:

$$\mathbb{E}_{\mathbf{x} \sim p_{\text{data}}(\mathbf{x}|z=k)}[S(\mathbf{x})] \approx \hat{\mu}_k(r) = \frac{\sum_t r_{t,k} S(x_t)}{\sum_t r_{t,k}} \, . \qquad (20)$$

Due to space constraints, we defer arguments for the following approximations to Appendix B:

$$\sum_t r_{t,k} \approx \sum_t s_{t,k} \approx \sum_t \tilde{s}_{t,k} = \sum_t \gamma_{t,k} \, , \qquad (21)$$

where the last equality comes from Eq. 17. Lastly, putting together Eqs. 18–20 and 21, we conclude

$$\nabla A(\boldsymbol{\phi}_k) \approx \hat{\mu}_k(\gamma) = \frac{\sum_t \gamma_{t,k} S(x_t)}{\sum_t \gamma_{t,k}} \, , \qquad (22)$$

demonstrating that our discriminative M-step is able to approximately match the stationary conditions of the exact M-step (Eq. 15). This is the essence of the discriminative M-step trick: we fit a discriminative model in the M-step in order to avoid dealing with the intractable normalizing constants, which turns out to approximate the stationary conditions of the exact M-step objective and facilitates efficient E-steps. Implementation details and pseudocode are provided in Appendix B.

### 3.4. Autoregressive Torus Graphs and Transfer Entropy Estimation

We now introduce autoregressive TGs (AR-TGs), which are a special case of conditional TGs where the covariates are lagged observations.

**Autoregressive Torus Graphs (AR-TG)** Let $X_t \in \mathbb{T}^d$ and $Y_t \in \mathbb{T}^1$ for $t \in \mathbb{N}$ be two time series of phase data and let $L \in \mathbb{N}$ be a lag. We write $Y_{<t} = Y_{t-1:t-L}$ and similarly for $X_{<t}$. An autoregressive torus graph (AR-TG) is a conditional TG in which the covariates are lagged observations:

$$p(y_t|\mathbf{x}_{<t}, y_{<t}) \propto \exp[\boldsymbol{\phi}(\mathbf{x}_{<t}, y_{<t})^\top S(y_t)] \qquad (23)$$

where we assume $\boldsymbol{\phi}$ is parameterized linearly in lag features:

$$\boldsymbol{\phi}(x_{<t}, y_{<t}) = \mathbf{b} + \sum_{\ell=1}^{L} \left( \mathbf{W}_\ell^{(y)} \, \psi(y_{t-\ell}) + \mathbf{W}_\ell^{(x)} \, \psi(\mathbf{x}_{t-\ell}) \right),$$
$$(24)$$

where $\psi(\theta) \triangleq [\cos\theta; \sin\theta]^\top$ embeds phases in $\mathbb{R}^2$, and is applied elementwise to vector arguments so that $\psi(\mathbf{x}) \in \mathbb{R}^{2d}$ for $\mathbf{x} \in \mathbb{T}^d$. This parameterization has three appealing properties: (i) it respects the periodicity of phase variables by embedding them via $\psi$ rather than using raw angles directly; (ii) it is the natural first-order approximation to

any smooth map from lagged phases to TG parameters; and (iii) it keeps the model statistically and computationally tractable, since the number of parameters scales linearly in $L$. More expressive parameterizations, such as deep neural networks applied to the lag features, are a natural extension for settings where the dependence on history is highly nonlinear.

**Bivariate Transfer Entropy Estimation**   A primary challenge in computational neuroscience is to infer *directional* interactions from observations alone. This is most often formulated as a transfer entropy estimation problem, with Granger causality being the most important instance (Schreiber, 2000; Barnett et al., 2009). Transfer entropy from time series $X$ to $Y$ is the reduction in the conditional entropy of $Y_t$ when the past of $X$ is included:

$$TE_{X \to Y} = \mathbb{H}(Y_t|Y_{<t}) - \mathbb{H}(Y_t|Y_{<t}, X_{<t}), \quad (25)$$

where $\mathbb{H}(\cdot)$ denotes differential entropy.

To estimate $TE_{X \to Y}$ we fit two AR-TG models on a training set: $\hat{p}_1(y_t \mid y_{<t})$ and $\hat{p}_2(y_t \mid y_{<t}, \mathbf{x}_{<t})$. On an independent test set $\{(\mathbf{x}_t, y_t)\}_{t=1}^T$ we form

$$\widehat{TE}_{X \to Y} = \frac{1}{T-L}\sum_{t=L+1}^{T}[\log \hat{p}_2(y_t|y_{<t}, \mathbf{x}_{<t}) \\ - \log \hat{p}_1(y_t|y_{<t})] \quad (26)$$

This is a difference of empirical cross-entropies. If the test set is independent of the training set and the model is well-specified $\widehat{TE}_{X \to Y}$ approaches $TE_{X \to Y}$ as $T \to \infty$. Because univariate TG models are von Mises-distributed, and therefore have tractable normalizing constants, we assume $y_t \in \mathbb{T}^1$ so that $\log \hat{p}_i(y_t|\cdot)$ is available in closed form. Consistency holds under standard assumptions (stationarity/ergodicity of the process, correct model specification, and an independent test set). We report TE in units of nats.

**Multivariate Transfer Entropy Estimation**   Calculating pairwise transfer entropies between $C$ channels using the above method involves estimating $\mathcal{O}(C^2)$ autoregressive models, which is impractical for large $C$. Therefore, we elect to estimate multivariate transfer entropy using a single estimated autoregressive model along with a flexible imputation model. Concretely, for three time series $X$, $Y$, and $Z$, we wish to estimate transfer entropies of the form

$$TE_{X \to Y|Z} \triangleq \mathbb{H}(Y_t|Y_{<t}, Z_{<t}) - \mathbb{H}(Y_t|Y_{<t}, X_{<t}, Z_{<t}). \quad (27)$$

We first train a full prediction model to estimate the autoregressive structure of the time series: $\hat{p}(X_t, Y_t, Z_t|X_{<t}, Y_{<t}, Z_{<t})$. As before, $\hat{p}$ factorizes as a product of one-dimensional von Mises conditionals. This forbids instantaneous causality by assuming conditional independence across units at time $t$ given the history.

Second, we train an imputation model which can be used to estimate the history of one time series from the rest, e.g. $\hat{p}(X_{<t}|Y_{<t}, Z_{<t})$. These two pieces allow us to form the necessary conditionals of the form $p(Y_t|X_{<t}, Y_{<t}, Z_{<t})$ and $p(Y_t|Y_{<t}, Z_{<t})$. Specifically, we approximate the latter terms as

$$p(Y_t|Y_{<t}, Z_{<t}) \approx \\ \mathbb{E}_{X_{<t} \sim \hat{p}(X_{<t}|Y_{<t}, Z_{<t})}[\hat{p}(Y_t|X_{<t}, Y_{<t}, Z_{<t})], \quad (28)$$

where the expectation is estimated by a small Monte Carlo sample. In practice we use a Gaussian model with full covariance as our imputation model, fit via MAP estimation with a small ridge prior to the cosine and sine features of windows of lagged observations, thereby embedding the phases into $\mathbb{R}^2$. As above, we learn a linear map from these cosine and sine features to the von Mises parameters.

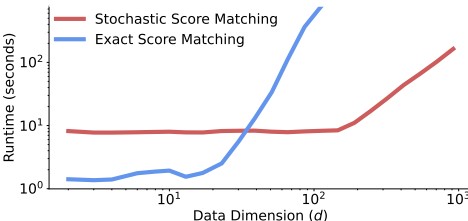

*Figure 2.* Runtime of stochastic versus exact score matching as a function of data dimension.

## 4. Results

We evaluated our methods on synthetic datasets to assess accuracy against known ground truth and on a large-scale mouse LFP dataset to illustrate practical utility. All experiments were run on a single Nvidia A5000 GPU with 24 GB VRAM using efficient JAX implementations (Bradbury et al., 2018). See Appendix C for additional experimental details. Code is available at https://github.com/jackgoffinet/torus-graphs.

### 4.1. Synthetic Data Experiments

We first tested whether stochastic score matching recovers ground-truth TG parameters. Using Hamiltonian Monte Carlo to sample from TGs with random parameters, we compared exact and stochastic estimation. Stochastic score matching matches the performance of exact methods for low-dimensional data and shows large gains in performance on higher-dimensional data (Figure 3A). Exact score matching performs well in low dimensions but fails at $d \approx 100$ due to memory demands (Figure 3B, Appendix C). In contrast, stochastic score matching operates at $d$ on the order of thousands and is an order of magnitude faster (Table 1).

Next, we asked whether TG-HMMs recover hidden states despite their approximate M-step. Simulated data show

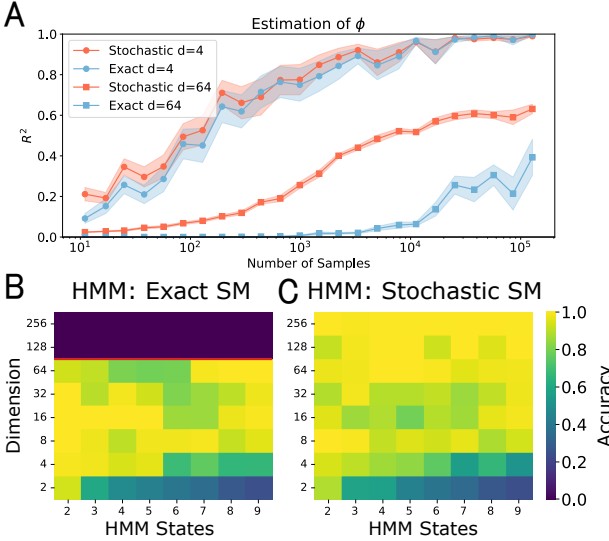

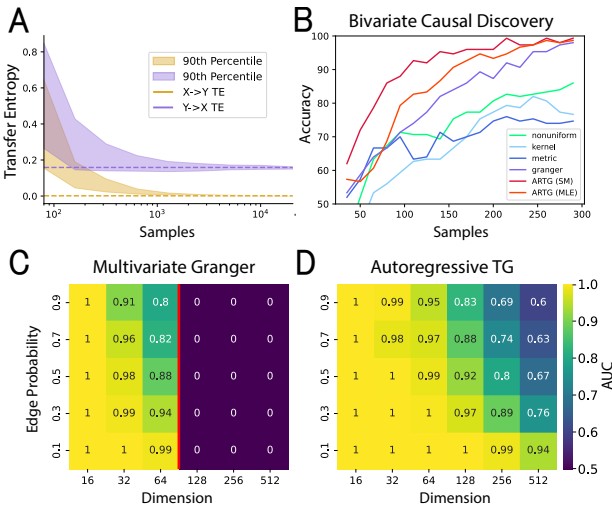

*Figure 3.* Synthetic validation of TG inference and scalability. **A:** Parameter recovery for TG models fit with increasing sample size in 4- and 64-dimensional synthetic data, measured by $R^2$ (squared Pearson correlation between ground-truth and estimated $\phi$ entries). Shading denotes $\pm$ SEM over 15 replicates. **B:** TG-HMM state recovery using exact score matching, measured by accuracy (fraction of timepoints assigned to the correct hidden state), which remains accurate up to $d \sim 100$ but exceeds GPU memory limits beyond this scale. **C:** Stochastic score matching matches exact performance at low dimensions and enables inference beyond $d \sim 100$ on the same hardware.

*Figure 4.* Transfer entropy (TE) estimation and causal discovery on synthetic data. **A:** Estimated TE from an autoregressive TG (AR-TG) converges rapidly to ground truth in a unidirectional setting. **B:** AR-TGs outperform baseline TE estimators in identifying causal direction. **C:** Multivariate Granger causality degrades rapidly with increasing dimension; the red line denotes a 30-hour timeout. **D:** AR-TGs maintain accurate and efficient causal edge recovery across a broader range of dimensions and sparsities.

good accuracy up to $d \sim 100$ for exact methods and well beyond that for stochastic methods (Figure 3B,C), demonstrating that the discriminative M-step approximation does not prevent accurate state recovery and that stochastic score matching enables inference in high-dimensional systems.

We then tested AR-TGs for transfer entropy estimation. In bivariate simulations with unidirectional influence, estimated TE converges quickly to ground truth (Figure 4A). Compared to a host of alternative TE estimators, AR-TGs more reliably identify the true direction of interaction (Figure 4B, see Appendix C for complete description of alternative methods). Interestingly, AR-TGs using score matching estimates outperform AR-TGs using maximum likelihood in this setting. Finally, in multivariate causal discovery tasks, AR-TGs maintain accurate edge recovery across a wider range of dimensions and sparsities than multivariate Granger causality (Figures 4C,D). Multivariate Granger hits a 30-hour timeout past $d = 64$, while each AR-TG run is under one hour.

In multivariate settings we employ transfer entropy estimation to perform causal discovery, classifying whether each potential directed edge between channels is active (high TE) or inactive (low TE). Compared to multivariate Granger causality (Figure 4c), the AR-TG is able to accurately per-

form causal estimation over a much wider range of channel counts (dimension) and sparsity levels (Figure 4d).

Overall, these synthetic experiments demonstrate that stochastic score matching enables accurate TG inference at scales where exact methods are infeasible, while preserving performance in dynamic and causal extensions. This motivates applying TGs, TG-HMMs, and AR-TGs to large real-world neural datasets.

### 4.2. Applications to Neural Data

We applied TG-based models to 48 hours of freely moving mouse LFP recordings from 62 channels with annotated sleep states. Phases were extracted via continuous wavelet transforms across 30 frequencies, yielding $d = 1860$ phase variables. This dimensionality is far beyond the scope of prior TG methods and serves as a stress test for scalable phase-only modeling on real neural data, enabling the extraction of structured phase relationships that may guide downstream scientific analysis.

#### 4.2.1. LARGE SCALE TGs

We fit separate TG models to Wake and NREM periods (Figure 5). The resulting phase statistics exhibit strong block structure across channel pairs and frequencies, while the learned TG parameters are substantially sparser, reflecting conditional independence after accounting for indirect interactions. Phase-sum terms are negligible, indicating that rel-

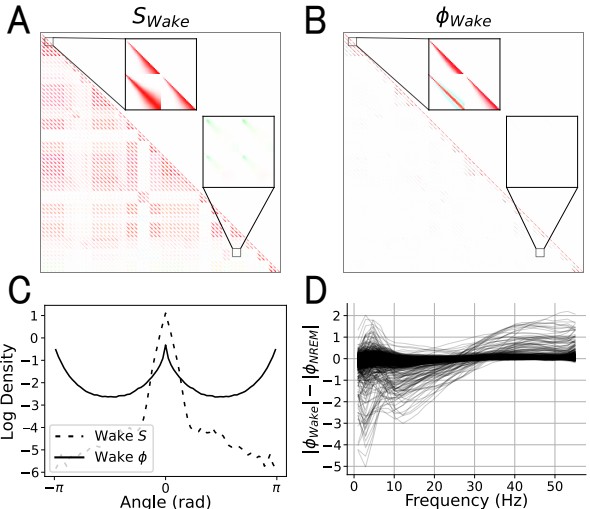

*Figure 5.* Large-scale TG analysis of mouse LFP phase data (62 channels, 30 frequencies, $d = 1860$). **A:** Empirical phase statistics arranged by channel–frequency pairs; insets show within- and between-region structure. Colors denote phase angle as in Figure 1. **B:** Corresponding TG parameters, revealing sparser direct interactions after accounting for conditional dependence. **C:** Distribution of phase angles in empirical statistics (dashed) and TG parameters (solid). **D:** Channel-wise differences in TG parameters between wake and NREM sleep.

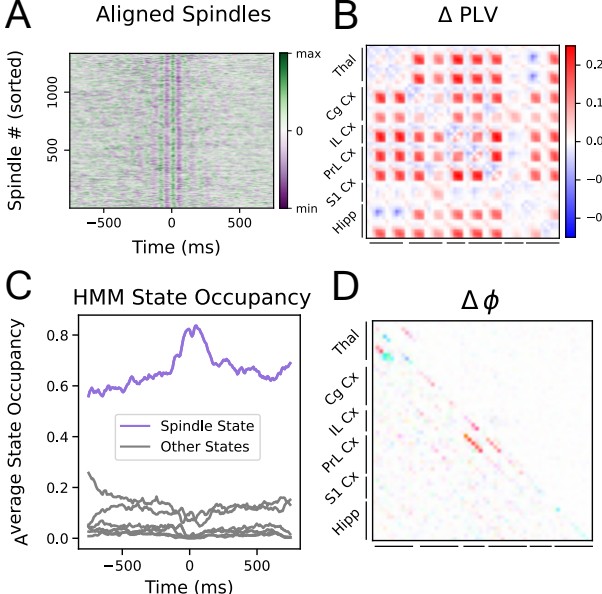

*Figure 6.* TG-HMM analysis of sleep spindles. **A:** Spindle waveforms from a representative channel, aligned to spindle center (0 ms). **B:** Change in phase-locking value (PLV) between near-spindle and far-from-spindle windows. **C:** State occupancy from a 6-state TG-HMM, highlighting a spindle-associated state. **D:** Difference in TG parameters between the spindle state and remaining states. Colors denote phase angle as in Figure 1.

ative phase differences dominate the structure. Comparing Wake and NREM models reveals a clear frequency-specific reorganization: high-frequency ($> 30$ Hz) couplings are stronger during Wake, whereas low-frequency ($< 30$ Hz) couplings dominate during NREM, consistent with established sleep physiology.

### 4.2.2. TG-HMM

To capture transient, state-dependent changes in phase coupling, we applied a TG-HMM to sleep spindles – brief $\sim$12 Hz oscillatory events during NREM (Figure 6). Across 1,334 aligned spindles, the TG-HMM inferred a small set of discrete states, one of which was strongly time-locked to spindle centers. This spindle-associated state exhibited sparse and interpretable changes in coupling structure, including enhanced cortical interactions and reduced thalamic high-frequency coupling. Notably, pairwise PLV analysis of the same spindle data shows diffuse, widespread synchrony increases (Figure 6B), whereas the TG-HMM reveals that the spindle-associated state is characterized by sparse, spatially specific changes in conditional coupling (Figure 6D). This contrast illustrates the added value of conditional independence modeling: the PLV increases largely reflect indirect interactions that are explained away once the full multivariate structure is accounted for.

### 4.2.3. AR-TG

We next analyzed directed interactions using autoregressive TGs with lag $L = 10$, combined with a Gaussian imputation model for multivariate transfer entropy (TE) estimation. TE patterns differed markedly between Wake and NREM and varied across frequency bands: beta-band ($\sim$18 Hz) interactions were generally stronger during NREM, while low-gamma ($\sim$45 Hz) interactions were stronger during Wake, indicating state-dependent routing of oscillatory influence. Multivariate TE further revealed prominent directed motifs across brain regions, including prelimbic cortex $\rightarrow$ striatum, infralimbic $\rightarrow$ prelimbic/cingulate cortices, and VTA $\rightarrow$ SNr, with distinct asymmetry profiles across behavioral states. These asymmetric directed interactions are not straightforward replications of existing connectivity findings, and suggest specific hypotheses about state-dependent information routing that can be tested in future experiments. Together, these results demonstrate that AR-TGs enable scalable, multivariate inference of directed phase interactions in large neural systems.

## 5. Discussion

We introduced scalable Torus Graph methods that make it possible to analyze phase relationships at previously inaccessible scales. By reducing the complexity of score

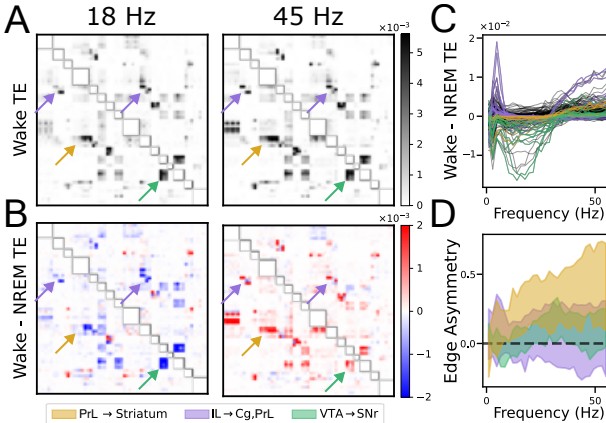

*Figure 7.* Multivariate transfer entropy (TE) in wake and NREM sleep. Connections of interest—prelimbic cortex → striatum, infralimbic → prelimbic/cingulate cortices, and VTA → SNr—are highlighted in yellow, purple, and green. **A:** TE between channels during wake at 18 and 45 Hz. **B:** Wake minus NREM TE at the same frequencies. **C:** Wake–NREM TE differences for all cross-region channel pairs across frequency. **D:** Minimum and maximum edge-asymmetry values for the highlighted connections, where edge asymmetry is defined as $(\widehat{TE}_{X \to Y|Z} - \widehat{TE}_{Y \to X|Z})/\max(\widehat{TE}_{X \to Y|Z}, \widehat{TE}_{Y \to X|Z})$, with values near $\pm 1$ indicating strongly asymmetric (directional) interactions and values near 0 indicating bidirectional interactions of similar magnitude.

matching from $\mathcal{O}(d^6)$ to $\mathcal{O}(d^2)$ per iteration, we extended TG inference from one hundred to nearly two thousand phase variables. Building on this foundation, we developed TG-HMMs to capture discrete, state-dependent changes in phase coupling and AR-TGs to infer directional dependencies through scalable transfer entropy estimation.

Our experiments on synthetic data demonstrate that these methods recover ground-truth structure and outperform standard alternatives. Applications to mouse LFPs allow us to see familiar neural data through a probabilistic phase-coupling lens, exposing interpretable structures that can guide future experimental study. While the frequency-specific reorganization between Wake and NREM broadly confirms known sleep physiology, and thus serves as a validation of the method, the sparse conditional structure revealed by the TG-HMM (Figure 6D) and the directed asymmetries identified by the AR-TG (Figure 7) are not straightforward replications of existing findings and suggest specific testable hypotheses about state-dependent neural activity.

Several limitations remain. First, the autoregressive approach to transfer entropy relies on univariate target variables, limiting its immediate application to multivariate outputs. Second, the discriminative M-step for TG-HMMs provides only approximate consistency, and its theoretical prop-

erties warrant further study. Third, while TG parameters provide a principled representation of conditional independence, their neuroscientific interpretation is less straightforward, which may hinder adoption. In particular, when phase variables span multiple frequencies, within-frequency cross-channel coupling and cross-frequency coupling are jointly modeled as homogeneous nodes, which may complicate interpretation of parameters corresponding to cross-frequency interactions. Restricting the model to within-frequency interactions is straightforward and may be preferable in settings where cross-frequency coupling is not of interest. Lastly, as with all Granger-style methods, the directed interactions inferred by AR-TGs reflect predictive rather than interventional causality, and should be interpreted accordingly.

Despite these limitations, the methods presented here provide a foundation for scalable, probabilistic modeling of circular-valued data. Future work could extend the discriminative M-step of the TG-HMM to other exponential families, develop scalable methods for TG marginalization, and explore circular extensions to other probabilistic models originally developed for Euclidean geometries.

## Acknowledgments

We thank Kafui Dzirasa and Kathryn Walder-Christensen for helpful discussions. We additionally thank Kafui Dzirasa for collecting and providing the LFP data used in this paper. Research reported in this publication was supported by the National Institute of Mental Health of the National Institutes of Health under Award Number R01MH125430. Research reported in this publication was also supported by the National Science Foundation through the Traineeship in the Advancement of Surgical Technologies, Award Number 2125528. The content is solely the responsibility of the authors and does not necessarily represent the official views of the National Institutes of Health or the National Science Foundation.

## Impact Statement

This paper presents work whose goal is to advance the field of Machine Learning. There are many potential societal consequences of our work, none of which we feel must be specifically highlighted here.

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

## A. Torus Graph Conditioning and Marginalization

Consider again the $d$-dimensional TG distribution with $d > 1$:

$$p(\mathbf{x}; \boldsymbol{\phi}) \propto \exp\left[\sum_{j=1}^{d} \phi_j^\top \begin{bmatrix} \cos x_j \\ \sin x_j \end{bmatrix} + \sum_{j<k} \phi_{jk}^\top \begin{bmatrix} \cos(x_j - x_k) \\ \sin(x_j - x_k) \\ \cos(x_j + x_k) \\ \sin(x_j + x_k) \end{bmatrix}\right] \tag{29}$$

**Conditioning**   Suppose we condition on $x_m = \theta$ for some $m \in [1, d]$. The result is a $(d-1)$-dimensional TG distribution with modified parameters $\phi'$. First, write the pairwise potential terms $\phi_{jk}$ as

$$\phi_{jk} = \begin{bmatrix} \alpha_{jk} \\ \beta_{jk} \\ \gamma_{jk} \\ \delta_{jk} \end{bmatrix} . \tag{30}$$

Note that we need to be careful about index ordering because $\phi_{jk}$ is only defined for $j < k$. For this reason, we define

$$(\tilde{\alpha}, \tilde{\beta}, \tilde{\gamma}, \tilde{\delta}) = \begin{cases} (\alpha_{mk}, \beta_{mk}, \gamma_{mk}, \delta_{mk}) & \text{if } m < k \\ (\alpha_{km}, -\beta_{km}, \gamma_{km}, \delta_{km}) & \text{otherwise} \end{cases} \tag{31}$$

to summarize the $m$-$k$ interactions, since $\sin(x_k - x_m) = -\sin(x_m - x_k)$, while all the other functions remain unchanged under index swaps. Lastly, we make use of the angle sum and difference identities

$$\cos(\theta - x_k) = \cos\theta\cos x_k + \sin\theta\sin x_k, \qquad \sin(\theta - x_k) = \sin\theta\cos x_k - \cos\theta\sin x_k$$
$$\cos(\theta + x_k) = \cos\theta\cos x_k - \sin\theta\sin x_k, \qquad \sin(\theta + x_k) = \sin\theta\cos x_k + \cos\theta\sin x_k$$

to calculate the effect of the observation $x_m = \theta$ on the univariate potential $\phi_k$. We conclude

$$\Delta\phi_k = \begin{bmatrix} (\tilde{\alpha} + \tilde{\gamma})\cos\theta + (\tilde{\beta} + \tilde{\delta})\sin\theta \\ (\tilde{\alpha} - \tilde{\gamma})\sin\theta + (\tilde{\delta} - \tilde{\beta})\cos\theta \end{bmatrix} . \tag{32}$$

In conclusion, let $\phi'$ be the parameters of $p(\mathbf{x}_{-m}|x_m = \theta)$. Then the **unary potentials** for each $k \neq m$ change as

$$\boxed{\phi'_k = \phi_k + \Delta\phi_k, \quad \Delta\phi_k = \begin{bmatrix} (\tilde{\alpha} + \tilde{\gamma})\cos\theta + (\tilde{\beta} + \tilde{\delta})\sin\theta \\ (\tilde{\alpha} - \tilde{\gamma})\sin\theta + (\tilde{\delta} - \tilde{\beta})\cos\theta \end{bmatrix}} . \tag{33}$$

All **binary potentials** remain unchanged:

$$\boxed{\phi'_{jk} = \phi_{jk} \text{ for } j < k, \ j \neq m, \ k \neq m} \tag{34}$$

**Marginalization**   In general, the TG family is *not* closed under marginalization; see Klein et al. (2020), Sec. 4.2, for a bivariate counterexample whose univariate marginals are not TG-distributed.

## B. TG-HMM details

### B.1. TG-HMM M-Step Approximation, Continued

We pick up the TG-HMM M-step derivation at Eq. 20:

$$\mathbb{E}_{\mathbf{x} \sim p_{\text{data}}(\mathbf{x}|z=k)}[S(\mathbf{x})] \approx \hat{\mu}_k(r) = \frac{\sum_t r_{t,k} S(x_t)}{\sum_t r_{t,k}} . \tag{35}$$

Now we assume emissions dominate the aggregate occupancy of the states:

$$\sum_t r_{t,k} \approx \sum_t s_{t,k} \tag{36}$$

Intuitively, we assume transitions can drive local sequence properties such as *when* a state switches, but time averages determine global sequence properties such as time averages. Next, we assume our free parameters $\{A_k\}$ approximate the true log partition functions $A(\boldsymbol{\phi}_k)$ up to an additive constant: $\forall k, \ A_k \approx A(\boldsymbol{\phi}_k) + c$ for some $c \in \mathbb{R}$. Below, we detail an initialization scheme that encourages this property. This implies

$$\sum_t s_{t,k} \approx \sum_t \tilde{s}_{t,k} \ . \tag{37}$$

Now, we make use of the first-order stationary condition of optimizing the $Q'$ objective (Eq. 17):

$$\sum_t \tilde{s}_{t,k} = \sum_t \gamma_{t,k} \ . \tag{38}$$

Combining Eqs. 36-38 gives:

$$\sum_t r_{t,k} \approx \sum_t \gamma_{t,k} \tag{39}$$

Now we expand the difference in moments under the true posterior $r$ and our surrogate emission model posterior $\gamma$:

$$\hat{\mu}_k(\gamma) - \hat{\mu}_k(r) = \underbrace{\frac{\sum_t (\gamma_{t,k} - r_{t,k}) S(x_t)}{\sum_t \gamma_{t,k}}}_{\text{term 1}} + \underbrace{\left( \frac{1}{\sum_t \gamma_{t,k}} - \frac{1}{\sum_t r_{t,k}} \right) \sum_t r_{t,k} S(x_t)}_{\text{term 2}} \tag{40}$$

By Eq. 39, both terms approximately cancel, assuming, in addition, that the posterior differences $\gamma_{t,k} - r_{t,k}$ have small weighted covariance with $S(x_t)$. This leaves us to conclude

$$\hat{\mu}_k(r) \approx \hat{\mu}_k(\gamma) \tag{41}$$

Lastly, putting together Eqs. 18-20 and 41, we conclude

$$\nabla A(\boldsymbol{\phi}_k) \approx \hat{\mu}_k(\gamma) = \frac{\sum_t \gamma_{t,k} S(x_t)}{\sum_t \gamma_{t,k}} \ , \tag{42}$$

demonstrating that our discriminative M-step is able to approximately match the stationary conditions of the exact M-step (Eq. 15). This is the essence of the discriminative M-step trick: we fit a discriminative model in the M-step in order to avoid dealing with the intractable normalizing constants, which turns out to approximate the stationary conditions of the exact M-step objective and facilitates efficient E-steps by avoiding intractable normalizers.

## B.2. TG-HMM Computational Complexity

Each EM iteration has three dominant computational components. First, the surrogate emission scores

$$\ell_{t,k} = \boldsymbol{\phi}_k^\top S(x_t) - A_k \tag{43}$$

must be evaluated for all timepoints and states, costing $\mathcal{O}(TKd^2)$. Second, the forward–backward algorithm and transition matrix update each cost $\mathcal{O}(TK^2)$. Third, the M-step re-estimates the $K$ TG parameter vectors; with stochastic score matching, each optimization step costs $\mathcal{O}(Kd^2)$ per iteration (Section 3.1), and the surrogate log-normalizer update is of the same order with minibatching. Suppressing fixed minibatch sizes and numbers of optimizer steps, the per-EM-iteration cost scales as $\mathcal{O}(TK^2 + TKd^2 + Kd^2)$. In the LFP experiments reported here, the emission score evaluations and the M-step re-estimates of the TG parameters dominate the forward–backward pass.

## B.3. TG-HMM Implementation Details

See Algorithm 1 for the TG-HMM pseudocode.

---

**Algorithm 1** Approximate EM for the TG-HMM

---

**Require:** Observations $\{x_t\}_{t=1}^T \subset \mathbb{T}^d$, number of states $K$, TG sufficient statistics $S(x)$, maximum iterations $M$

1: Initialize TG parameters $\{\phi_k\}_{k=1}^K$, surrogate log normalizers $\{A_k\}_{k=1}^K$, uniform initial probabilities, and transition matrix $\Pi$

2: **for** $m = 1, \ldots, M$ **do**

3:     **E-step**

4:     Set surrogate log emissions $\ell_{t,k} \leftarrow \phi_k^\top S(x_t) - A_k$

5:     Run forward–backward using $\ell_{t,k}$ and $\Pi$ to obtain $\gamma_{t,k}$ and $\xi_{t,i,j}$

6:     **M-step: transition update**

7:     $\Pi_{ij} \leftarrow \dfrac{\sum_{t=2}^T \xi_{t,i,j} + \alpha_{ij} - 1}{\sum_{j'=1}^K \left(\sum_{t=2}^T \xi_{t,i,j'} + \alpha_{ij'} - 1\right)}$

8:     **M-step: surrogate log-normalizer update**

9:     $\{A_k\}_{k=1}^K \leftarrow \arg\min\limits_{\{A_k\}} -\sum_{t=1}^T \sum_{k=1}^K \gamma_{t,k} \log \dfrac{\exp(\phi_k^\top S(x_t) - A_k)}{\sum_{j=1}^K \exp(\phi_j^\top S(x_t) - A_j)} + \lambda_A \sum_{k=1}^K A_k^2$

10:     **M-step: TG parameter update**

11:     **for** $k = 1, \ldots, K$ **do**

12:        $\phi_k \leftarrow \arg\min\limits_{\phi} \sum_{t=1}^T \gamma_{t,k} \mathcal{L}_{\mathrm{SM}}(x_t; \phi) + \lambda_{\mathrm{TG}} \|\phi\|_2^2$

13:     **end for**

14: **end for**

15: **return** $\{\phi_k\}_{k=1}^K, \{A_k\}_{k=1}^K, \Pi$

---

**Initialization** We initialized the $K$ $\phi_k$ parameters by first fitting a base $\phi_{\mathrm{base}}$ to all the observations using stochastic score matching (2000 iterations, batch size 512, learning rate 0.01, $L_2$ regularization $\lambda = 0.1$). We next drew 100 very small perturbations of $\phi_{\mathrm{base}}$: $\phi_{\mathrm{init}}^{(i)} = \phi_{\mathrm{base}} + \epsilon$ where $\epsilon \sim \mathcal{N}(0, 10^{-4}I)$. Each timepoint was assigned the $\phi_{\mathrm{init}}$ that had the lowest energy (TG log probability, assuming the same normalizing constant of 1 for each TG). A co-assignment similarity matrix (fraction of shared labels across draws) was clustered with spectral clustering ($K$ clusters), producing hard pseudo-labels. We then refit one TG per cluster with the same stochastic score matching settings as above, yielding the initial set of TG parameters $\{\phi_k\}_{k=1}^K$. Before EM, we initialized the free log partition parameters $\{A_k\}_{k=1}^K$ by minimizing the cross entropy of the cluster assignments given the TG softmax scores with Adam for 2000 steps. The loss included an $L_2$ regularization on $\log Z_k$: $\lambda_Z \sum_k \|\log Z_k\|_2^2$ with $\lambda_z = 10^{-2}$. Note that the use of small perturbations from a single base $\phi$ encourages the $K$ initial TG parameters to have similar log normalizing constants, which helps ensure the approximation in Eq. 37, at least at initialization.

**Stability Measures** During EM, we use a short temperature warmup on emission log-scores that linearly scales a temperature parameter $\tau$ from $\tau_{\mathrm{init}} = 20$ over $n_{\mathrm{warmup}} = 20$ EM iterations. We also included a decaying teacher-forcing term that mixes the pre-EM pseudo-posteriors into the emissions with a linearly decaying weight from 1 to 0 over the $n_{\mathrm{warmup}}$ warmup iterations. TG parameters were re-estimated at each iteration using either exact score matching (with $L_2$ regularization $\lambda_{TG} = 0.1$) or by stochastic score matching (lr=$10^{-2}$, $L_2$ regularization $\lambda = 0.1$, 500 iterations, patience= 100, determined by exponential moving average loss with $\alpha = 0.99$). Exponential smoothing was applied to the TG parameters: $\phi^{(n)} \leftarrow \beta_n \phi^{(n)} + (1 - \beta_n)\phi^{(n-1)}$. Here, $\beta_n$ decays linearly from 1 to $\beta_{\mathrm{min}} = 0.1$ over the total number of EM iterations.

**State Priors** The initial state probabilities were uniform $\pi_k = \frac{1}{K}$ and were not updated over EM iterations. Transition probabilities were initialized from a sticky Dirichlet prior with self-transition weight $\alpha_{\mathrm{self}} = 10$ and off-diagonal weight $\alpha_{\mathrm{other}} = 1$, yielding a slightly self-biased matrix. These priors were retained as pseudo-counts in the M-step transition matrix update.

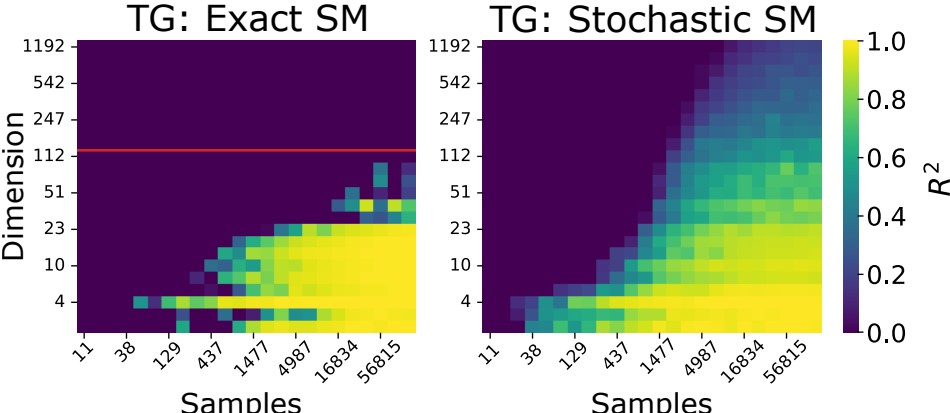

*Figure 8.* Additional Experiments comparing torus graph models fit using stochastic and exact methods. Pseudo-$R^2$ of the estimation of torus graph parameters ($\phi$) fit using an increasing number of samples from a synthetic time series from 2- to 1192-dimensional data. Stochastic score matching achieves comparable performance in low-dimensional settings, as well as achieving vastly better estimation in higher-dimensional settings. The red line represents the data dimension at which exact methods run out of memory (¿24 GB VRAM)

## C. Experimental Details

### C.1. TG: Stochastic vs. Exact Score Matching

**Synthetic Data**   We first sampled TG parameters $\phi$ by drawing entries i.i.d. from a standard normal distribution. Half of the matrices' off-diagonal elements were set to 0 at random to simulate a non-fully connected torus graph. We then sampled from the distribution using Hamiltonian Monte Carlo, as implemented in blackjax (Cabezas et al., 2024), with a step size of $3 \times 10^{-2}$ and 60 integration steps to ensure sufficient mixing.

**Model Fitting**   These samples were used to infer an estimate of the true TG parameters $\phi$ using both exact score matching and stochastic score matching. For both exact and stochastic score matching, we applied no regularization. For stochastic score matching, we ran the Adam optimizer for 5000 iterations using a batch size of 128 and a learning rate of $3 \times 10^{-3}$. Results are reported as an $R^2$ value defined as the Pearson Correlation Coefficient $R$ of $\phi_k$ (the ground truth parameters) and $\hat{\phi}_k$ (the estimated parameters) squared, averaged over 3 replicates.

**Additional Experiment**   Additional results are available in Figure 8. The red line shows when exact methods run out of memory (>24 GB VRAM) on an Nvidia A5000 GPU, around $d = 120$. Stochastic methods do not run out of memory on the same hardware even on data with up to thousands of dimensions. Stochastic methods also show a more graceful curve of performance degradation compared to exact methods, which easily become unstable. Performance is reported as a pseudo-$R^2$ value that assumes unit variance (equivalent to 1 – normalized MSE over the $d^2$ parameters) and is bounded to the non-negative case: $\max \left(1 - \frac{\sum_k (\phi_k - \hat{\phi}_k)^2}{d^2}, 0\right)$, where $\phi_k$ denotes the ground-truth TG parameters and $\hat{\phi}_k$ denotes the estimated TG parameters for the $k^{\text{th}}$ experimental replicate and $d$ is the dimension of the TG observations. To construct the heatmap, these pseudo-$R^2$ values were averaged over three replicates.

### C.2. TG-HMM: Stochastic vs. Exact Score Matching

**Synthetic Data**   We start by sampling TG parameters $\{\phi_k\}_{k=1}^K$, where K is the number of ground truth HMM states, by drawing $\phi_k$ i.i.d. from a normal distribution with the variance of $\phi$'s diagonal terms equal to 1 and the variance of off-diagonal terms equal to 2. Each row of the HMM transition matrix was sampled from the Dirichlet distribution with $\alpha_{\text{self}} = 10$ and $\alpha_{\text{other}} = 1$ to encourage the creation of a transition matrix that has a relatively higher chance of not making a transition at any timepoint. This more accurately reflects real-world data, where states usually persist longer than one time step. We then sample states from the transition matrix and observations from the TG corresponding to each timepoint's state, using Hamiltonian Monte Carlo, as above, to sample from the TG.

**Model Fitting**    These samples were used to estimate the true TG parameters for each state $\phi_k$, as well as the state assignment of each timepoint using both exact and stochastic score matching. The TG-HMM is fit using 50 EM iterations with a learning rate of $10^{-2}$ and 20 warmup iterations, where the temperature linearly decays from 2 down to 1. To encourage more gradual updates over the EM process, exponential smoothing was applied to the TG parameters: $\phi^{(n)} \leftarrow \beta\phi^{(n)} + (1 - \beta)\phi^{(n-1)}$ with $\beta = 0.5$. In each of the 50 iterations, we performed 200 iterations of Adam to update the log partition function to help ensure convergence. The accuracy of the estimated latent states after the final EM iteration is reported. Accuracies in the heatmap were averaged over three replicates.

### C.3. Bivariate Causal Discovery

**Synthetic Data**    We simulated a bivariate causal system where there is a causal link from $Y \to X$ but no such link from $X \to Y$. We simulate this system to allow us to estimate a "causal direction" without having to determine the magnitude of causality, which becomes difficult to compare between different causal discovery methods. This system was simulated by first creating a linear mapping matrix $W$ to describe the causal relationship between the variables and within the variables for each lag up to 10 (as parameterized in Eq. 24, with $\mathbf{b} = \mathbf{0}$). The variance of each entry in this $W$ matrix was set to $e^{-0.9*(10-\ell)}$ where $\ell$ is the lag, except for the entries describing the $X \to Y$ connection, which were set to 0 to enforce a zero transfer entropy from $X$ to $Y$. Note that $W$ is also structured to predict products of univariate von Mises distributions, which corresponds to a TG distribution with no angle sum or angle difference terms. To kickstart the sampling process, a 10-timepoint initial random history of uniform angles was generated. For each subsequent timepoint, the von Mises statistics are calculated autoregressively and the next angles are sampled from the resulting von Mises distributions.

**Model Fitting**    Each transfer entropy method was used to estimate the transfer entropy from $X \to Y$ and $Y \to X$ given the generated samples (the TG method used raw angle values while the others used the cosine of the angles).

The other methods used for comparison are

- The Python package IDTxl's (Wollstadt et al., 2018) implementation of a Kraskov (nonuniform) transfer entropy estimator

- The Python package infomeasure's (Büth et al., 2025) implementation of kernel density estimation transfer entropy estimation

- The Python package infomeasure's (Büth et al., 2025) implementation of Kozachenko-Leonenko (KL) / Metric / kNN transfer entropy estimation

- The Python package statsmodels' (Seabold et al., 2010) implementation of Granger causality tests

For all methods, a ground truth lag of 10 was given before the model was fit. All methods used recommended parameters to emulate best practice. We used the default parameters for the statsmodels and IDTxl methods, and for the infomeasure methods we used the parameters given in the "Demos" section of their documentation.

We repeated the sampling and modeling procedure 150 times for each sample size shown in Figure 4b to average over variability introduced by drawing a random system on each run. Figure 4b reports the resulting "accuracy" as a function of the number of samples. Here, accuracy denotes the fraction of resamplings for which the estimated transfer entropy from $Y \to X$ exceeded that from $X \to Y$, with the additional requirement that $\widehat{TE}_{Y \to X} > 0$. These conditions ensure the methods both infer the correct causal direction ($\widehat{TE}_{Y \to X} > \widehat{TE}_{X \to Y}$) and detect genuine causal influence ($\widehat{TE}_{Y \to X} > 0$).

**Convergence Plot**    In Figure 4A, the data was sampled in the same way as above. 40 random seeds were run and the space between the 95th and 5th percentiles was filled in to smooth out variance and remove outliers. The "ground truth" lines shown on the plot were determined by averaging five estimations of the transfer entropy using 100,000 simulated samples.

## C.4. Multivariate Causal Discovery

**Synthetic Data** We first randomly generate the sparsity pattern of the linear mapping matrix $W$ (from Eq. 24 with $\mathbf{b} = \mathbf{0}$), where each pair of signals is coupled with probability equal to the specified edge probability. The nonzero entries of $W$ (corresponding to coupled signal pairs) and all diagonal entries are then sampled independently from a standard normal distribution. All remaining entries are set to zero. Samples are drawn using the same procedure described above, extended to a multivariate setting with a lag of 8.

**Model Fitting** We fit both an AR-TG model for transfer entropy estimation and a multivariate Granger model to compare their ability to recover the causal structure of the data. For the Granger baseline, a multivariate vector autoregressive (VAR) model from the statsmodels package was fit to the cosine-transformed phase variables. Granger causality tests were then performed for each pair of variables, and the resulting $F$-statistics were used to compute AUC values. The AR-TG model used a multivariate normal (MVN) distribution to impute missing channels in the history, with the MVN mean and covariance estimated from all available data. The covariance was regularized by adding $10^{-1}I$. Model parameters were optimized by minimizing the negative log-likelihood for 4,000 iterations using a learning rate of $3 \times 10^{-3}$ and a batch size of 64, without additional regularization (e.g. $L_2$ regularization). Transfer entropy between each unique pair of channels (conditioned on the remaining channels) was estimated using four Monte Carlo samples per prediction and 3,200 total predictions.

## C.5. LFP Dataset Details

The mouse LFP dataset consists of 62 LFP channels and a single trapezius EMG channel recorded in a home cage environment continuously for 48 hours at a sampling rate of 1 kHz. We downsampled each signal to 250 Hz because we only considered frequencies up to 55 Hz, well below the Nyquist frequency of the downsampled signal. Along with this dataset is a collection of sleep state labels, classifying each 2-second window as Wake, NREM, or REM sleep based on the power of the EMG signal and the low-frequency content of a cortical channel.

**Extracting Phase via Continuous Wavelet Transforms** To extract phases, we first calculated a continuous wavelet transform (CWT) of each channel and each desired frequency independently (Morlet wavelet, width parameter 5). We then took the complex angle to associate each channel, frequency, and timepoint with a phase.

### C.5.1. SPINDLE DATASET DETAILS

We extracted putative sleep spindle times from LFPs using a single prelimbic cortex reference channel, and then extracted 2-second multichannel windows centered at each event. The reference signal was zero-phase band-passed (Butterworth, order 4) from 9 to 16 Hz. We formed the Hilbert envelope, Gaussian-smoothed it ($\sigma = 75$ ms), and robust-z scored it (median/median absolute deviation). Candidate events were found with two-sided thresholds ($1 < z < 3$) and pruned by duration (0.5–1.5 s) with merging for gaps less than 250 ms. To ensure oscillatory content, we required median instantaneous frequency (Hilbert phase derivative) within 9–16 Hz and $\geq 3$ cycles per event. Artifacts were vetoed if high-frequency (40–100 Hz) or low-frequency (0.5–4 Hz) envelopes exceeded 3.5 z inside the interval. Events were centered at the envelope peak, and 2-s windows from all channels were saved. A second pass refined timing. For each spindle, we selected an integer shift ($\pm 50$ ms) maximizing cosine similarity to a leave-one-out mean template of the reference-channel waveform. The shift was then applied to all channels, and a 1.5-s window centered at each event was saved.

## C.6. TG LFP Application Details

To estimate the average TG statistics across the 62 channels and 30 frequencies in Figure 5a, we averaged the empirical statistics over a random set of 512,000 Wake-labeled timepoints. To estimate the TG parameters in Figure 5b (and the corresponding NREM parameters), we ran stochastic score matching for 12,000 iterations with batch size 32, learning rate $3 \times 10^{-3}$, and $L_2$ regularization $\lambda = 0.1$. We observed that 12,000 iterations was enough for the training loss to plateau.

Larger versions of Figures 5A and B, with brain region labels, are provided in Figures 9 and 10.

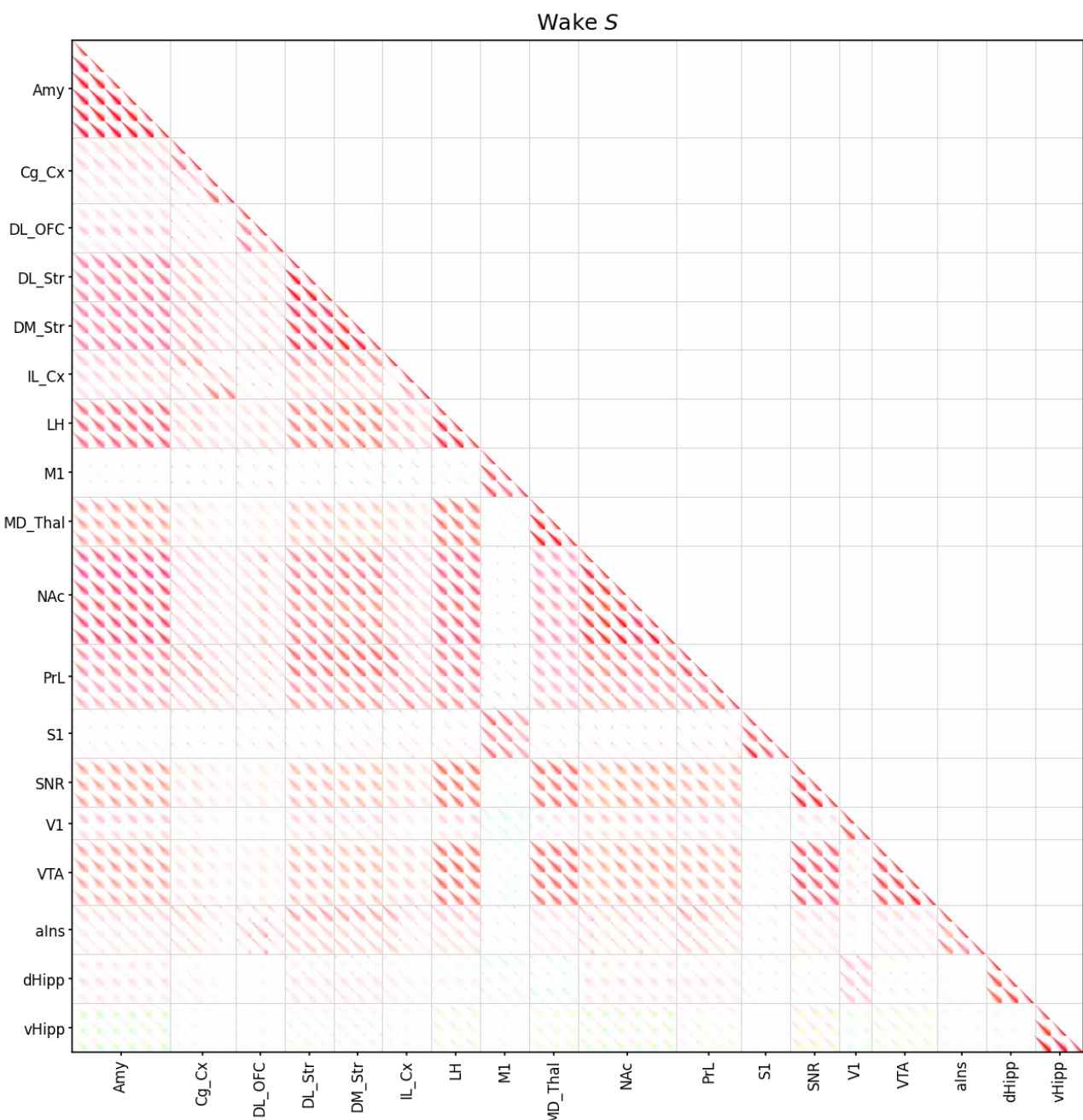

*Figure 9.* Wake TG statistics with brain region labels. Rows and columns are grouped by brain region (18 regions, 2–6 channels each), and each channel contributes 30 phase variables at center frequencies linearly spaced between 1 and 55 Hz.

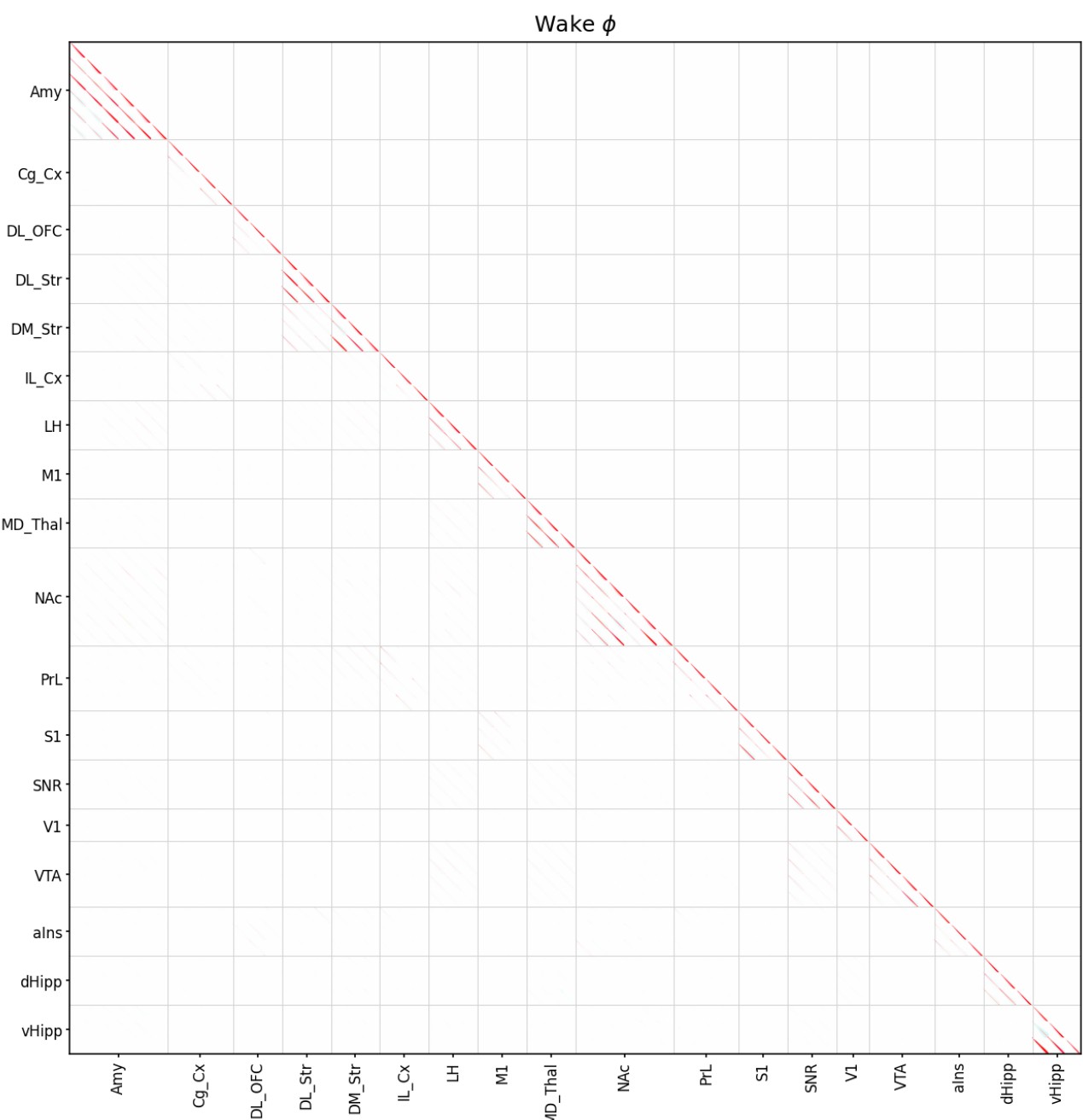

*Figure 10.* Wake TG parameters with brain region labels.

| SM Method | $d = 16$ | $d = 64$ | $d = 1024$ |
|---|---|---|---|
| Exact | $607 \pm 3$ | $1143 \pm 2$ | Out of Memory |
| Stochastic | $\mathbf{137 \pm 3}$ | $\mathbf{139 \pm 5}$ | $\mathbf{310 \pm 2}$ |

*Table 1.* Mean $\pm$ std. dev. runtime of each method, in seconds, corresponding to runs in Figure 2.

### C.7. TG-HMM Application Details

To fit the TG-HMM to sleep spindle data (Figure 6), we first collected 1,334 putative sleep spindles using the method described above. We then trained a $K = 6$ state TG-HMM on the first 100 spindles (37,500 total timepoints, 40 EM iterations, $n_{\text{warmup}} = 30$, $\tau_{\text{init}} = 20$, stochastic score matching regularizations $\lambda_{L_2} = 10^{-1}$ and $\lambda_{L_1} = 3 \times 10^{-2}$). The trained model was then applied to all 1,334 sleep spindles to calculate the average state occupancies in Figure 6c. Mean spindle waveforms for all channels are shown in Figure 11.

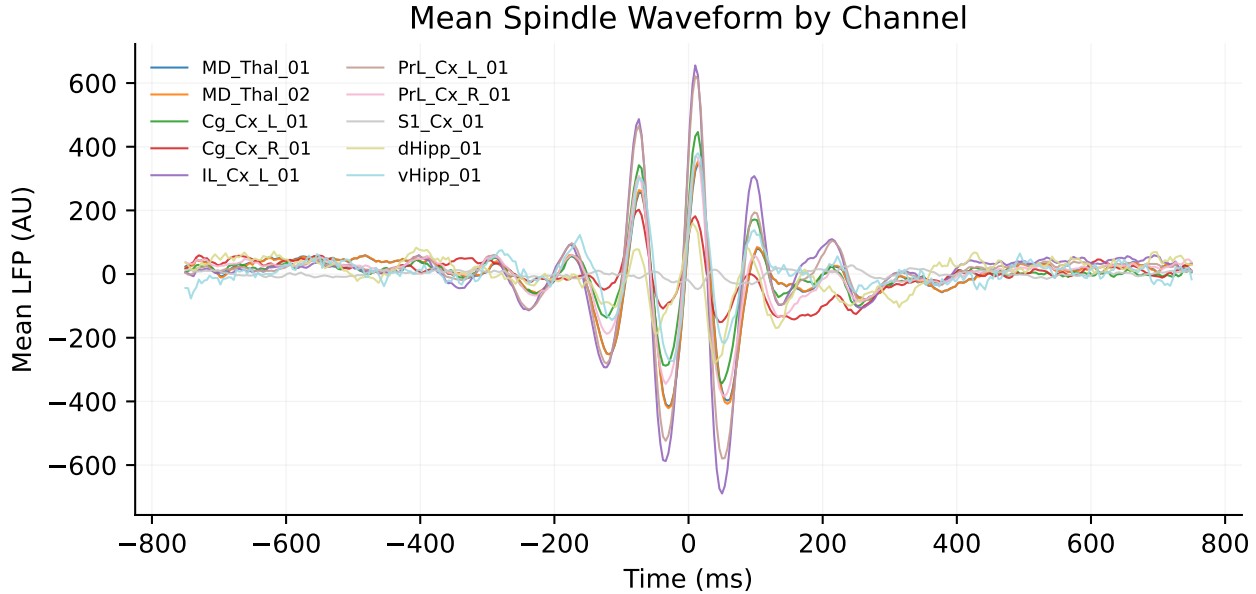

*Figure 11.* Mean spindle waveforms for each channel in Figure 6.

### C.8. AR-TG Application Details

**Imputation Model**   We used a multivariate normal (MVN) distribution to impute missing channels from the history. With 62 channels, 10 lagged timesteps ($L = 10$), 30 frequencies per channel, and a cosine and sine embedding of each angle, the resulting imputation model is a 37,200-dimensional MVN. We estimated the mean and covariance of this MVN using the first two moments of 1,344,000 random windows of phases (observations). Lastly, we regularized the covariance by adding $10^{-1} I$.

**AR-TG Training**   We trained the AR-TG using a negative log-likelihood loss for 40,000 iterations, a learning rate of $3 \times 10^{-3}$, and a batch size of 64 without additional regularization (e.g. $L_2$ regularization).

**TE Estimation**   We estimated multivariate TE (between each pair of unique channels, conditioned on the remaining 60 channels) using 8 Monte Carlo samples per prediction (approximating the expectation in Eq. 28) and 512,000 total random samples.

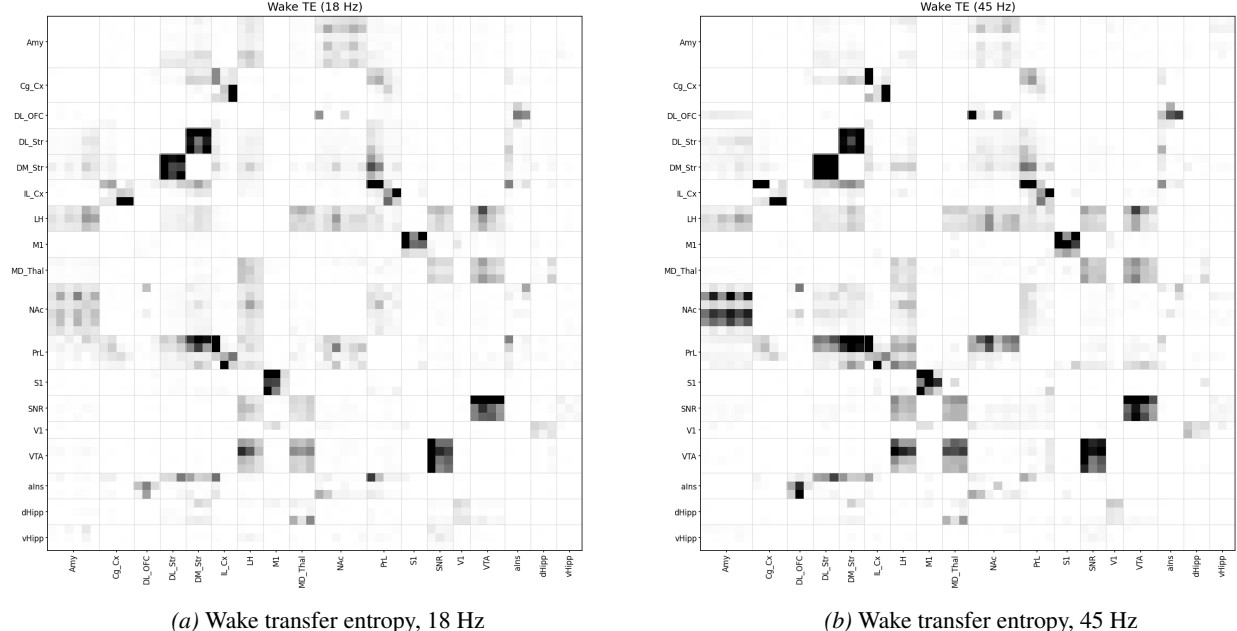

*(a)* Wake transfer entropy, 18 Hz    *(b)* Wake transfer entropy, 45 Hz

*Figure 12.* Large labeled version of Figure 7A.

## C.9. Model Specification Stability and Sensitivity Analysis

We additionally report sensitivity analyses examining the robustness of each method to hyperparameter choices and model specification.

**Stochastic Score Matching TG Sensitivity to Hyperparameter Specification**    To investigate the sensitivity of SSM-TG to its hyperparameters we conducted line searches over batch size, learning rate, and optimizer, utilizing 5 random initializations for each on synthetic data with a ground truth $\phi$. Within any given hyperparameter setting, the average Pearson correlation of estimated $\phi$ across seeds is $0.998$, demonstrating that random initialization does not produce spurious structure. Across batch sizes, inter-setting correlation is $0.991$. Higher learning rates (`1e-3` to `7e-3`) performed best (correlation to true $\phi$: $0.884$); with solutions resulting from Adam and AdamW being essentially identical. This shows that SSM-TG is not sensitive to hyperparameter choices within a reasonable range. In addition, we observed stable and monotonic convergence of the score-matching objective across runs, with consistent stopping based on validation loss. Plots of the training loss are for these models are available in figure 14

**TG-HMM Sensitivity to Number of States (K)**    In addition to the six state model in Figure 6C for the HMM sleep spindle task we also fit models with $K \in [3, 8]$. In these models we were also able to identify a spindle locked state with an occupancy that peaks near the spindle time, recapitulating the key finding in Figure 6C. See Figure 15 for state occupancies relative to the spindle time for all trained models.

**Surrogate Controls and Lag Sensitivity for AR-TG**    We conducted a negative control by training an AR-TG on 16 synthetic time series (ground truth lag 8), then rolling one time series by 31 timesteps (exceeding the ground truth lag) to break its causal structure. Estimated TE from the rolled series to others dropped by $61.9\%$ on average ($0.111 \rightarrow 0.042$), while TE among all other pairs was essentially unchanged (correlation: $0.991$). We emphasize that, as with all transfer entropy methods, this reflects predictive (Granger-style) causality rather than interventional causality.

We additionally conducted lag sensitivity analysis: in a bivariate system with ground truth lag 10, estimated TE from the causal direction decreases only slowly as the specified lag of the AR-TG increases from 10 to 30 ($0.306 \rightarrow 0.282$), while TE in the non-causal direction remains near zero throughout. This shows that AR-TG is not sensitive to moderate misspecification of lag order.

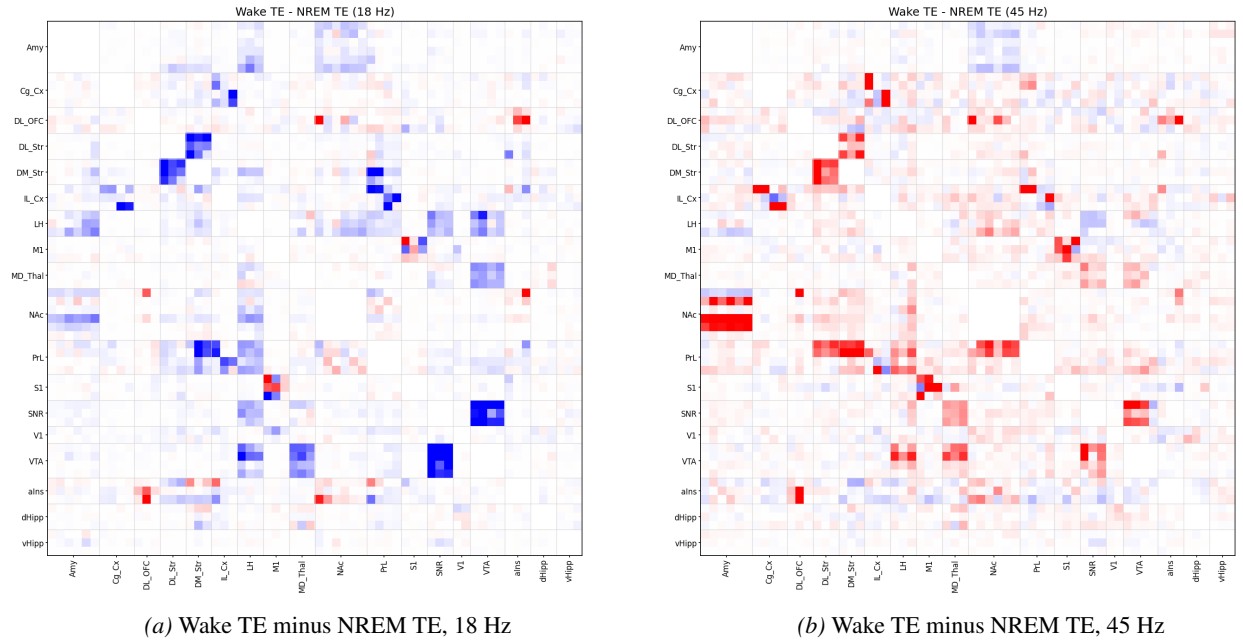

*(a)* Wake TE minus NREM TE, 18 Hz

*(b)* Wake TE minus NREM TE, 45 Hz

*Figure 13.* Large labeled version of Figure 7B.

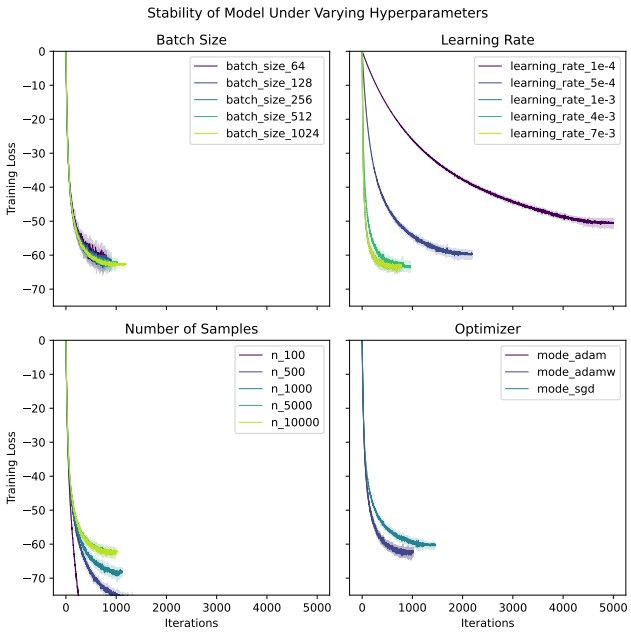

*Figure 14.* Training Loss curves used in stability analysis. Mean over 5 runs is plotted ± standard deviation. Stopping criteria was applied independently for each model.

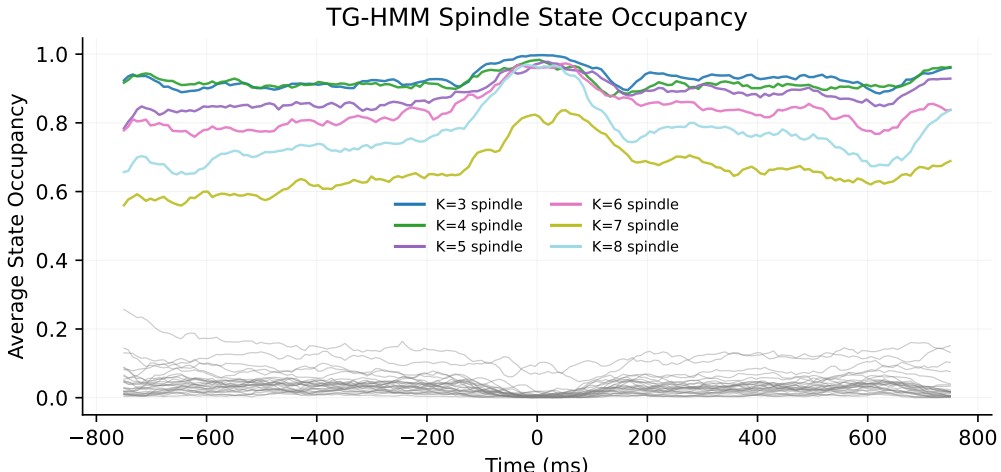

*Figure 15.* State occupancies relative to spindle times for all $K$-state TG-HMMs with $K \in [3, 8]$. Compare to Figure 6C.

