# OpenReview forum: "Torus Graphs for Large Scale Neural Phase Analysis"
_ICML.cc/2026/Conference — ICML 2026 regular_

### Official Review · Reviewer_ZYb8 · 2026-02-28

**Soundness:** 3
**Presentation:** 2
**Significance:** 2
**Originality:** 2
**Overall Recommendation:** 3
**Confidence:** 4

**Summary:**

This paper proposes scalable Torus Graphs for large-scale neural phase-coupling analysis. It replaces the original TG fitting procedure based on score matching with O(d6) complexity by a stochastic score-matching approach, reducing the per-iteration cost to 𝑂(𝑑2) and the memory requirement from O(d4) to 𝑂(𝑑2), thereby enabling modeling of phase variables in the thousand-dimensional regime. Building on this foundation, it further extends TG to TG-HMM to capture state-dependent changes in phase coupling, and introduces an autoregressive TG framework combined with transfer entropy to estimate directional interactions. Experiments on both synthetic data and 48-hour mouse LFP recordings demonstrate the ability to handle 1,860 phase features derived from 62 channels and 30 frequency bands, revealing frequency-specific reorganization of coupling between wakefulness and NREM sleep, transient spindle-related states, and directed information-flow patterns across brain regions.

**Compliance With Llm Reviewing Policy:**

Affirmed.

**Key Questions For Authors:**

When phase variables from channel by frequency-band combinations are jointly modeled in a high-dimensional framework, within-frequency coupling and cross-frequency coupling differ in their physical interpretation, yet the manuscript does not sufficiently describe the variable construction and the boundaries of interpretability. It remains unclear whether cross-frequency phase relationships should be treated equivalently to within-frequency phase synchrony, and modeling all frequency-specific phases as homogeneous nodes could introduce difficult-to-interpret or potentially spurious edges. I recommend explicitly defining the modeling target, distinguishing whether the analysis focuses on within-frequency cross-channel coupling only or also includes cross-frequency relationships, and providing ablation comparisons that justify the chosen representation and demonstrate its interpretability and robustness.

**Limitations:**

yes

**Strengths And Weaknesses:**

The manuscript emphasizes that modern recordings have reached a high-dimensional scale on the order of hundreds of channels by hundreds of frequency bands, and argues that traditional circular statistics or pairwise measures are inadequate. However, the notion of a scaling gap is not quantified with sufficient precision. The authors state that classical circular statistics are limited to only a few variables, that phase-locking value is restricted to pairwise analysis, and that high-dimensional structure remains relatively unexplored. It would be important to clarify at approximately what dimensionality existing approaches become impractical in practice. In particular, it is unclear whether the dominant limitation arises from statistical considerations such as estimator variance under limited sample sizes, or from computational and memory constraints. Without a clearer characterization of the failure boundary, the contribution may be perceived as addressing an overstated bottleneck. I recommend adding typical dataset scales and a summary of the dimensional ranges that mainstream methods can handle, supported by appropriate citations, to strengthen the motivation in the backgrd.

The background justification for why multivariate modeling based on conditional independence is necessary, rather than relying on pairwise phase-locking value or coherence, remains largely conceptual. The manuscript notes that empirical analyses often rely on amplitude-based measures or pairwise phase-based measures, and argues that these are insufficient to reveal high-dimensional multivariate phase-only structure. It also emphasizes that the proposed TG framework enables multivariate conditional independence structure inference, which is not achievable with pairwise approaches. However, the background does not include an intuitive example illustrating how pairwise synchrony can induce spurious connections due to indirect pathways or common input, and how conditional independence analysis can mitigate these effects. Without such an example, readers may consider a pipeline based on pairwise estimation and subsequent graph construction to be an adequate alternative, thereby weakening the motivation. I recommend adding a simple illustrative example or citing established neuroscience evidence that demonstrates the scientific necessity of multivariate conditional independence analysis over pairwise measures.

The manuscript uses stochastic score matching to reduce the computational complexity of TG to quadratic scaling in the dimensionality, but does not provide a systematic analysis of gradient variance and convergence stability. Stochastic estimators can be sensitive to choices such as batch size, learning rate, and sampling strategy, and suboptimal settings can lead to slow convergence or high variability in the learned structure. If the paper reports only final performance metrics without training dynamics or variability across repeated runs, it is difficult to assess the robustness and stability of the optimization procedure. I recommend including convergence curves and reporting confidence intervals or variability over multiple random seeds under different batch and sampling configurations, along with a clearly stated stopping criterion.

The number of parameters in TG increases rapidly with dimensionality, reflecting the involvement of pairwise interactions that scale quadratically in the number of variables, which raises concerns about overfitting in neural datasets with limited recording duration. The manuscript does not sufficiently discuss the use of sparsity, low-rank structure, or explicit regularization, nor does it explain how related hyperparameters are selected. Without a sensitivity analysis for regularization strength and model selection, the inferred network structure may reflect noise rather than genuine coupling. I recommend specifying the regularization terms used, the hyperparameter search ranges, and providing ablation studies that evaluate the stability of recovered structures under different regularization settings.

TG-HMM represents temporal variability through a finite number of latent states, but the manuscript does not adequately describe the trade-off between the choice of the number of states and interpretability. The selected number of states can substantially affect segmentation granularity, transition dynamics, and consequently downstream neuroscientific conclusions. If the manuscript relies on a single state number or selects it solely via validation performance, the resulting interpretation may be statistically optimized yet not robust. I recommend reporting sensitivity analyses over multiple state numbers, such as three, five, seven, and ten states, and demonstrating whether key inferred structures and conclusions remain consistent across these choices.

The combination of AR-TG with transfer entropy is used to study directionality and information flow, but transfer entropy estimation is often highly sensitive to assumptions of stationarity, the choice of lag order, and unobserved confounding. Without explicit controls for common drivers or volume conduction, transfer entropy findings may be misinterpreted as evidence of causal influence. In addition, autoregressive modeling depends strongly on lag order selection and regularization, and the Methods section would benefit from clearer justification of these choices. I recommend including negative control analyses using time-shifted or surrogate data, reporting sensitivity to lag order, and providing a discussion of confounding, with an explicit clarification that the inferred directionality should be interpreted as predictive rather than interventional causality.

When phase variables from channel by frequency-band combinations are jointly modeled in a high-dimensional framework, within-frequency coupling and cross-frequency coupling differ in their physical interpretation, yet the manuscript does not sufficiently describe the variable construction and the boundaries of interpretability. It remains unclear whether cross-frequency phase relationships should be treated equivalently to within-frequency phase synchrony, and modeling all frequency-specific phases as homogeneous nodes could introduce difficult-to-interpret or potentially spurious edges. I recommend explicitly defining the modeling target, distinguishing whether the analysis focuses on within-frequency cross-channel coupling only or also includes cross-frequency relationships, and providing ablation comparisons that justify the chosen representation and demonstrate its interpretability and robustness.

---

> ### Author Rebuttal · Authors · 2026-03-31
>
> We thank the reviewer for their detailed and constructive critique. We address each concern in turn.
>
> **Necessity of conditional independence over pairwise measures:** Klein et al. (2020) provide a concrete neuroscience demonstration of exactly this point (their Figure 7): PLV indicates synchrony between dentate gyrus (DG) and prefrontal cortex (PFC), but TG reveals that this is spurious -- DG and PFC are conditionally independent given subiculum, which acts as an intermediary. More generally, pairwise measures cannot distinguish direct from indirect interactions, so common inputs or multi-hop pathways can induce spurious edges in pairwise connectivity graphs. We will mention this example in the main text to strengthen the motivation.
>
> **Dimensionality failure boundary:** PLV is restricted to pairwise comparisons and does not provide conditional independence structure, regardless of dimensionality. Exact TG inference runs out of GPU memory at $d \approx 128$ (> 24 GB VRAM on an A5000). Our stochastic score matching operates at $d=1860$ on the same hardware. Multivariate Granger causality hits a 30-hour timeout past $d=64$ (Figure 4C), whereas the AR-TG scales to 512 (Figure 4D). More broadly, the failure modes differ across methods: pairwise measures are statistically limited (they cannot recover conditional structure), while multivariate models such as TG and Granger causality are primarily limited by computational and memory constraints, with additional statistical challenges (e.g., variance and overfitting) arising at high dimensionality. Our work specifically addresses the computational bottleneck, enabling exploration of regimes where these statistical questions become meaningful.
>
> **Gradient variance and convergence stability:** We conducted line searches over batch size, learning rate, and optimizer, each with 5 random seeds on synthetic data with known ground-truth $\phi$. Within any given hyperparameter setting, the average Pearson correlation of estimated $\phi$ across seeds is 0.998, demonstrating that random initialization does not produce spurious structure. Across batch sizes, inter-setting correlation is 0.991. Higher learning rates (1e-3 to 7e-3) performed best (correlation to true $\phi$: 0.884); Adam and AdamW were essentially identical. SSM-TG is not sensitive to hyperparameter choices within a reasonable range. In addition, we observed stable and monotonic convergence of the score-matching objective across runs, with consistent stopping based on validation loss. We will include convergence curves and variability across seeds in the revision.
>
> **TG-HMM sensitivity to K:** We ran the spindle TG-HMM for $K \in [3,9]$. A spindle-locked state appears consistently across all values of $K$. The number of states affects granularity of the remaining states but does not alter the key finding. We will add this analysis to the paper.
>
> **Surrogate controls for AR-TG transfer entropy:** We conducted a negative control by training an AR-TG on 16 synthetic time series (ground truth lag 8), then rolling one time series by 31 timesteps (exceeding the ground truth lag) to break its causal structure. Estimated TE from the rolled series to others dropped by 61.9% on average (0.111 $\rightarrow$ 0.042), while TE among all other pairs was essentially unchanged (correlation 0.991). We emphasize that, as with all transfer entropy methods, this reflects predictive (Granger-style) causality rather than interventional causality. We also conducted a lag sensitivity analysis: in a bivariate system with ground truth lag 10, estimated TE from the causal direction decreases only slowly as lag increases from 10 to 30 (0.306 $\rightarrow$ 0.282), while TE in the non-causal direction remains near zero throughout. AR-TG is not sensitive to moderate misspecification of lag order. We will add these analyses to the Appendix and will apply an analogous phase-randomization control to the LFP TE estimates in the revision.
>
> **Cross-frequency vs. within-frequency coupling:** We agree this distinction is worth flagging for readers. Our current approach treats all phase variables as homogeneous nodes, which captures cross-frequency as well as within-frequency cross-channel coupling. We view this as the most parsimonious modeling choice. Restricting to within-frequency interactions only would require discarding potentially meaningful cross-frequency structure. That said, we acknowledge that cross-frequency TG parameters require more careful neuroscientific interpretation, and we will add a clarifying discussion of this point, including the note that the framework trivially supports within-frequency-only analyses by zeroing the relevant parameters.
>
> **Regularization:** We use L2 regularization throughout (regularization strengths stated in the appendix). Our framework also supports group-L1 regularization. We will consolidate regularization details into the main text and report sensitivity to regularization strength in the revision.

---

> > ### Author Rebuttal · Reviewer_ZYb8 · 2026-04-02
> >
> > I thank the authors for their thoughtful and detailed rebuttal. My major concerns have been adequately addressed. I appreciate the clarifications and additional analyses provided in response to the review, which have strengthened the paper. I hope the reviewer comments will help the authors further improve the manuscript, both in terms of presentation and overall technical merit.

---

### Official Review · Reviewer_yWRU · 2026-03-11

**Soundness:** 3
**Presentation:** 3
**Significance:** 3
**Originality:** 3
**Overall Recommendation:** 4
**Confidence:** 3

**Summary:**

This paper studies scalable probabilistic modeling of multivariate neural phase data on the torus. Building on Torus Graphs (TG), the authors propose a stochastic score-matching procedure that reduces the inference cost from \(O(d^6)\) to \(O(d^2)\), enabling modeling of datasets with thousands of phase variables. Based on this scalable foundation, the paper further introduces two extensions: (i) TG-HMM for state-dependent phase coupling and (ii) AR-TG for estimating directional interactions via transfer entropy. Experiments on synthetic data and mouse LFP recordings demonstrate the ability to recover large-scale phase interaction patterns and state-dependent coupling structures.

**Compliance With Llm Reviewing Policy:**

Affirmed.

**Final Justification:**

Thank you for the detailed clarification. My concerns have been adequately addressed.

**Key Questions For Authors:**

N/A

**Strengths And Weaknesses:**

**Strengths**
- Addresses an important problem: large-scale multivariate modeling of neural phase interactions beyond pairwise coupling statistics.
- The framework is extended to temporal and directional settings (TG-HMM and AR-TG), increasing its applicability.
- Experiments include both synthetic validation and real LFP data, demonstrating feasibility at large scale.


**Weaknesses**
- Real-data experiments mainly demonstrate feasibility at scale, but the neuroscientific interpretation is limited. The paper identifies state-dependent phase coupling. Is there any close connection of these findings to neurophysiological mechanisms, nor does it show whether the results provide novel biological insight beyond confirming known phenomena.

---

> ### Author Rebuttal · Authors · 2026-03-31
>
> We thank the reviewer for their positive assessment. We briefly address the concern about neuroscientific novelty.
>
> The reviewer notes that some findings (stronger high-frequency coupling during Wake and low-frequency coupling during NREM) confirm known neurophysiology. We agree, and view this as a form of validation: a new method that contradicted well-established sleep physiology would be cause for concern. Beyond this sanity check, we would point to findings that are less anticipated. The TG-HMM analysis of sleep spindles reveals that the spindle-associated state is characterized by sparse, spatially specific changes in conditional coupling, whereas pairwise PLV analysis of the same data shows diffuse, widespread synchrony increases. This contrast is not a replication of prior work. It reflects the added value of modeling conditional independence rather than marginal synchrony. Similarly, the AR-TG directed motifs (e.g., prelimbic cortex $\rightarrow$ striatum, VTA $\rightarrow$ SNr asymmetries across behavioral states) are not straightforward replications of existing findings and suggest specific hypotheses about state-dependent information routing that can be tested in future experiments. We will foreground these points more clearly in the revision.

---

> > ### Author Rebuttal · Reviewer_yWRU · 2026-04-01
> >
> > Thank you for the detailed clarification. My concerns have been adequately addressed.

---

### Official Review · Reviewer_h6ud · 2026-03-12

**Soundness:** 3
**Presentation:** 1
**Significance:** 3
**Originality:** 3
**Overall Recommendation:** 5
**Confidence:** 4

**Summary:**

The manuscript studies Torus Graph (TG) models for analyzing phase relationships in time series, motivated by applications in neuroscience where neural signals are often represented by their instantaneous phases. The authors build on the existing TG distribution (Klein et al 2020 AoAS) and focus on improving the computational scalability of parameter estimation, which becomes prohibitive in high-dimensional settings. The paper proposes a more efficient estimation approach that substantially reduces the computational burden relative to earlier methods, enabling the model to be applied to larger systems. Leveraging this improvement, the paper further introduces extensions such as a Torus Graph Hidden Markov Model for modeling time-varying phase interactions and an autoregressive TG framework for directional analysis. The proposed methods are evaluated through simulation studies and applications to neural recordings, which suggest that the framework can capture meaningful patterns of phase coupling.

**Compliance With Llm Reviewing Policy:**

Affirmed.

**Final Justification:**

All concerns raised during the review and the discussion period have been adequately addressed. I maintain my score: Accept.

**Key Questions For Authors:**

1. *Comparison with existing scalable TG estimation methods* - Previous work (e.g., Klein et al. 2026 AoAS) proposes alternative approaches (such as sparsity inducing regularization) for reducing the computational burden of TG estimation. Did the authors consider these methods, and how does the proposed approach compare empirically?

2. *Clarification of evaluation metrics* - Several figures report performance metrics (e.g., Parameter Recovery $R^2$), but these metrics are not formally defined in the manuscript. Could the authors clearly define all evaluation measures and explain what is being plotted in Figures 3–7?

3. *TG-HMM estimation procedure* - The description of the TG-HMM parameter estimation in Section 3.3 is difficult to follow. Could the authors provide a clearer explanation of the algorithm and its computational complexity?

4. *Model specification choices* - In the autoregressive TG formulation, the target variable $Y_t$ restricted to a one-dimensional torus. What challenges will appear if the target is extended to higher-dimensional toroidal variables?

5. *Loss of amplitude information* - The analysis focuses exclusively on phase variables. Under what conditions is ignoring the amplitude of the original signals justified, and when might this lead to significant loss of information?

6. *Citation needed* - In Section 4.1, are the authors referring to the method by Klein et al. 2020 AoAS by the term 'Exact score matching'?

7. Could the authors justify the linear parameterization of TG parameters in equation 24?

Apart from the above, there are minor typographical errors, e.g., see the use of $h(x)$ in equations 8 and 9.

**Limitations:**

Yes.

**Strengths And Weaknesses:**

**Strengths**

1. *Strong theoretical foundation* - The TG model is built on a well-established statistical framework for circular variables developed by Klwin et al 2020 AoAS. It provides a principled probabilistic model for phase interactions.

2. *Significant computational contribution* - The primary contribution of the paper is a new parameter estimation strategy that substantially reduces the computational cost compared with previous approaches. This makes the TG models more feasible for high dimensional systems.

3. *Methodological extensions* - The work develops extensions of the TG framework (e.g., TG-HMM and autoregressive TG models), which become computationally tractable due to the proposed estimation method.

4. *Empirical validation* - Simulation studies and real data analyses demonstrates promising performance of the proposed approach.

5. *Potential relevance for neuroscience applications* - The framework is well motivated by problems involving phase relationships in neural oscillations.



--------------------------------------------------------------
**Weaknesses**


1. *Clarity and presentation issues* - Several parts of the manuscript are difficult to follow due to missing definitions, inconsistent notation, and insufficient explanations. In particular, the description of the TG-HMM estimation procedure (Section 3.3) is hard to follow.

2. *Undefined or poorly introduced concepts* - Several technical terms appear throughout without definition or citation (e.g., complex angle, continuous wavelet transform in defining the visualization matrix in Page 3 ), which may confuse readers who are not already familiar with the domain.

3. *Incomplete descriptions of figures and metrics* - The figures are often difficult to interpret because the performance metrics are not defined, axes are not clearly labeled, and the captions do not adequately explain what is being shown. This affects several figures (e.g., Figures 3–5).

4. *Limited discussion of modeling assumptions* - The analysis relies solely on phase information, ignoring signal amplitude, but the manuscript does not sufficiently discuss when this loss of information may be problematic.

5. *Connection to existing literature could be clearer* - Some related methods for scalable TG estimation (e.g., Hamiltonian Montel Carlo, conditional-density approaches and exact score matching methods) are mentioned but not discussed, or not properly cited.

6. *Accessibility issues*- The paper assumes considerable domain knowledge; several acronyms and technical concepts are introduced without explanation, reducing accessibility for a broader machine learning or statistics audience. For example, Section 4.2 requires the reader to have substantial domain knowledge in order to comprehend the results.

---

> ### Author Rebuttal · Authors · 2026-03-31
>
> e thank the reviewer for their careful and constructive reading. We address each question in turn.
>
> **Klein et al.'s sparsity-inducing regularization:** We agree that Klein et al. discuss group $\ell_1$-regularized score matching as a way to encourage sparse graph structure in high dimensions. This is complementary to our contribution. Their approach adds sparsity to the TG objective, whereas our focus is the separate bottleneck of making TG estimation itself scalable in very high dimensions. In particular, they do not provide a scalable optimization procedure analogous to our stochastic score-matching approach. Our framework is compatible with both $L_2$ and group-$\ell_1$ regularization, so these ideas could be combined. We also note that methods such as Hamiltonian Monte Carlo are designed for sampling rather than direct parameter estimation, and therefore do not address the same scalability challenge for fitting TG models in high dimensions.
>
> **Figure clarity and undefined metrics:** We acknowledge this is a presentation weakness and will address it in the revision. Specifically: Figure 3A reports Pearson $R^2$ between ground-truth and estimated $\phi$ entries, averaged over replicates. We will add a definition in the caption. Figures 3B and 3C report state-recovery accuracy (fraction of timepoints assigned to the correct hidden state). Again, we will define this explicitly. In Figures 5–7 (LFP results) we will clarify that complex phase colors correspond to the colorwheel in Figure 1. We will present expanded versions of 5A, 5B, 7A, and 7B in the Appendix to show the names of all channels. Colorbars will be added for 6A and 6B. "Edge Asymmetry" will be defined in the caption of Figure 7.
>
> **Undefined terms:** We will add brief definitions of "continuous wavelet transform" and "complex angle" on first use, and expand the caption of Figure 1 to explain the visualization matrix construction without assuming prior familiarity.
>
> **AR-TG extension to higher-dimensional targets:** When the target variable $y_t$ lies on a higher-dimensional torus $\mathbb{T}^d$ for $d > 1$, the univariate conditional distributions are no longer von Mises but rather more general TG distributions, which have intractable normalizing constants. Estimating transfer entropy requires evaluating log-likelihoods of the form $\log p(y_t \mid \cdot)$, which is tractable in closed form only when $y_t \in \mathbb{T}^1$ (the von Mises case). For $d > 1$, one would need to approximate the normalizing constant, which would complicate the TE estimator. This is why we restrict to $y_t \in \mathbb{T}^1$ in this work. We will add this explanation to the limitations section.
>
> **The amplitude assumption:** Phase reduction theory provides the primary theoretical justification: when oscillatory dynamics lie near a stable limit cycle, amplitude perturbations decay rapidly and phase becomes the primary degree of freedom governing long-term behavior. The hardware motivation is complementary: amplitude is often more sensitive than phase to electrode referencing and frequency-dependent recording gain. We would also note that analyzing amplitude or power while discarding phase is itself a common and accepted choice in the neuroscience literature. Our work takes the deliberate opposite stance, focusing entirely on phase while setting aside amplitude. This is an intentional modeling choice that enables a clean probabilistic framework, not an oversight, though we agree it should be stated more clearly in the paper.
>
> **The linear parameterization in Eq. 24:** The linear parameterization is the natural first-order approximation: it assumes that the TG natural parameters vary smoothly with the lagged phase history, embeds phases into $\mathbb{R}^2$ via cosine and sine features, and maps these to parameters via a learned linear transformation. This choice keeps the model statistically and computationally tractable while respecting circular structure, and performed well in our experiments. Higher-order (e.g., neural network) parameterizations are a natural extension for settings where the dependence on history is highly nonlinear, and we will note this explicitly.
>
> **"Exact score matching" terminology:** Yes, this refers to the closed-form linear solve proposed by Klein et al. (2020, AoAS). We will clarify this on first use.
>
> **TG-HMM estimation procedure:** We agree that the description in Section 3.3 can be clarified and will revise it for readability, including a more explicit step-by-step summary of the algorithm. We will also explicitly state the computational complexity: each EM iteration consists of a forward–backward pass with cost $\mathcal{O}(T K^2)$, followed by $K$ TG re-estimation steps; with stochastic score matching, each of these scales approximately as $\mathcal{O}(d^2)$ per optimization step.

---

> > ### Author Rebuttal · Reviewer_h6ud · 2026-04-01
> >
> > I thank the authors for the rebuttal. My major concerns have been adequately addressed. I hope the comments from the reviewer will help the authors significantly improve their manuscript both in terms of presentation and merit.

---

### Official Review · Reviewer_gPFz · 2026-03-14

**Soundness:** 2
**Presentation:** 3
**Significance:** 2
**Originality:** 3
**Overall Recommendation:** 5
**Confidence:** 4

**Summary:**

This work builds upon the Torus Graph (TG) model, an exponential-family distribution defined on phase variables that can represent conditional dependencies among oscillations. The authors propose a stochastic score matching approach that reduces the per-iteration computational cost to $O(d^2)$, enabling scalable inference for large TG models. Experiments on LFP recordings demonstrate that the proposed approach can model high-dimensional phase interactions and reveal dynamic phase-coupling structures across brain states such as wakefulness and sleep.

**Compliance With Llm Reviewing Policy:**

Affirmed.

**Final Justification:**

The authors' rebuttal addresses my concerns. I hope they will include some suggestions into the final version to further clarify paper's positioning.

**Key Questions For Authors:**

My main concern is about the necessity of using torus graph for neural signal analysis.
The paper is motivated by EEG and LFP data, but it is not very clear why these signals particularly require torus graph modeling. For EEG or sEEG data, spatial relationships between electrodes are usually very important, and many existing works model these spatial structures explicitly. For LFP signals, the analysis often focuses more on local activity or key features from specific brain regions, rather than global interactions. Therefore, it would be helpful if the authors could further clarify why torus graph is the most suitable framework for these neural signals, and what advantages it has compared with existing approaches.

**Limitations:**

Please refer to weakness and questions.

**Strengths And Weaknesses:**

- Strength:
1) The paper studies an important research question of the modeling phase interactions in neural oscillatory data such as EEG and LFP. Understanding such phase relationships across many channels is an important topic in neuroscience, and existing approaches often rely on pairwise metrics such as coherence or PLV, which cannot capture higher-order dependency structures.
2) A key contribution of the paper is a stochastic score matching approach that significantly improves the computational scalability of torus graph inference.
3) The paper writing is easy to follow.

- Weakness:
1) While the paper is motivated by neural recordings such as EEG, it is not very clear why the proposed method is particularly suitable for EEG analysis. For example, many EEG studies consider spatial relationships among channels, frequency bands, or other neurophysiological priors. However, these aspects are not explicitly modeled in this work. In addition, there are many graph learning methods designed for EEG representation learning or clinical/BCI tasks, but the paper does not clearly discuss how the proposed torus graph method differs from or relates to these approaches.
2) The experiments mainly focus on demonstrating the scalability of torus graph models and their ability to discover phase coupling structures. However, the paper does not include comparisons with commonly used EEG analysis methods, such as spectral connectivity measures [1], graph-based models [2], or other modern multivariate EEG modeling approaches [3]. Including these baselines would help better understand the advantages of the proposed method.
3) The main contribution of the paper is an efficient inference algorithm for torus graph models. The modeling framework itself is not new, and the main improvement is making the existing model scalable to larger neural datasets.
4) More ablation studies would be helpful. For example, it would be useful to analyze how the proposed method behaves under different noise levels, signal conditions, or hyperparameter settings.

-----

[1] Self-Supervised Graph Neural Networks for Improved Electroencephalographic Seizure Analysis, ICLR 2022.

[2] EvoBrain: Dynamic Multi-Channel EEG Graph Modeling for Time-Evolving Brain Networks. NeurIPS 2025.

[3] EEG2Rep: Enhancing Self-supervised EEG Representation Through Informative Masked Inputs, KDD 2024.

---

> ### Author Rebuttal · Authors · 2026-03-31
>
> We thank the reviewer for their thoughtful comments. We agree that the paper would benefit from clearer positioning relative to the broader EEG literature, and we will revise the manuscript to make the intended use case more explicit.
>
> **Why torus graphs for neural phase data:** Our goal is not to model all aspects of EEG/LFP data, nor to maximize performance on downstream clinical or BCI tasks. Rather, we focus on a specific scientific question: multivariate phase-only dependence. In this setting, torus graphs are attractive because they define a probabilistic model directly on circular variables, and their pairwise interaction parameters encode conditional dependence structure among phases. This allows us to move beyond pairwise metrics such as PLV or coherence and ask which phase relationships remain after accounting for indirect effects from all other variables.
>
> We agree that spatial priors among electrodes are often valuable in EEG analysis. However, incorporating such priors is orthogonal to the main contributions of this paper, which are: (i) a scalable inference procedure for TGs, (ii) a TG-HMM for dynamic state-dependent phase coupling, and (iii) an AR-TG framework for directional phase interactions via transfer-entropy estimation. These contributions would remain applicable whether or not one adds anatomical or spatial regularization.
>
> **Relation to cited EEG graph-learning methods:** The cited works address different objectives from ours. The ICLR 2022 seizure-GNN paper and EvoBrain are discriminative graph-based models for seizure detection / prediction, while EEG2Rep is a self-supervised representation-learning method for EEG tasks. These are important and relevant adjacent works, but they are not probabilistic graphical models over circular phase variables, and their learned graph structures are not directly interpretable as TG-style conditional-dependence structure. We will revise the paper to clarify this distinction and cite these methods as complementary rather than directly competing baselines.
>
> **Baselines:** We agree that stronger discussion of baseline choice would help. For the inference problem we study, the most direct baselines are pairwise phase-connectivity measures (e.g., PLV/coherence) and exact TG inference. The former are appropriate because they are standard tools for phase analysis, and the latter isolates the effect of our scalable inference procedure on the same modeling family. We will expand the discussion of baseline choice to make this reasoning explicit.
>
> **Originality:** While the TG family itself is prior work, we believe the contribution is broader than an engineering speedup. First, the stochastic score-matching formulation exploits the structure of the TG sufficient statistics to avoid forming the dense score-matching system, making inference practical at much larger scales. Second, this scalability enables two additional methodological extensions in the paper: the TG-HMM and the AR-TG framework. We will revise the manuscript to better separate these contributions and clarify the scope of the novelty claim.

---

> > ### Author Rebuttal · Reviewer_gPFz · 2026-04-03
> >
> > Thank you for the clarification. I now have a much clearer understanding of the work and have increased my rating to 3. I still have two remaining questions:
> > 1) I would like to better understand the motivation behind incorporating phase information in EEG analysis. While related prior work may exist, it would be very helpful if the authors could provide concrete case studies or illustrative examples (e.g., real EEG signals) to demonstrate why phase analysis is necessary and beneficial.
> > 2) While the method and experiments show the effectiveness of the torus graph, the **necessity of using atorus graph** for EEG analysis remains unclear. In particular, the technical motivation behind this design choice is not sufficiently justified. As mentioned by the authors (e.g., line 72), the motivation appears to be largely investigative, which makes the work seems like an empirical study. A clearer explanation of why the torus graph is specifically required, rather than alternative representations, would strengthen the contribution.
> >
> > If the authors can provide stronger justification for these two points, I will further increase my rating for finalisation.

---

> > > ### Author Response · Authors · 2026-04-07
> > >
> > > Thank you for your continued engagement and for the updated rating.
> > >
> > > **Motivation for phase-based analysis in EEG:** We agree that this should be made more concrete. Phase specifies where an ongoing EEG oscillation is within its cycle at a given moment, which can be relevant both for local excitability and for coordination across regions. As one concrete EEG example, Busch et al. (2009) showed that whether a near-threshold visual stimulus is detected depends in part on the prestimulus phase of ongoing EEG oscillations. This illustrates that phase in scalp EEG can reflect behaviorally relevant moment-to-moment changes in brain state, and therefore motivates phase-based analysis alongside power-based approaches. We will add this example to the introduction to make the motivation more intuitive.
> > >
> > > **Why TGs for neural phase analysis:** Torus graphs are particularly well-suited to the problem we study. Our goal is to model multivariate dependence among circular (phase) variables in a way that supports interpretable conditional relationships and efficient inference. Torus graphs provide a minimal probabilistic framework that satisfies these requirements.
> > >
> > > 1) TGs are defined directly on the torus, respecting the periodic nature of phase variables.
> > > 2) TGs model multivariate conditional independence structures rather than only pairwise relationships.
> > > 3) TGs admit practical inference by score matching, made more practical by the proposed stochastic score matching (SSM) method.
> > >
> > > We are not aware of alternative approaches that jointly satisfy these requirements. For example, restricting Euclidean models to $[0,2\pi)^d$ and interpreting each dimension as a phase variable introduces an artificial discontinuity at $0$, contradicting property 1. Phase locking value (PLV; Lachaux et al. 1999) only characterizes bivariate systems and therefore does not admit a conditional independence interpretation. Lastly, other possibilities like wrapped Gaussian models or kernel density estimate-based models do not admit practical inference.
> > >
> > > Together, these properties make TGs the natural modeling choice for this problem. We will clarify this positioning in the revision.
> > >
> > > * Busch, N. A., Dubois, J., & VanRullen, R. (2009). The phase of ongoing EEG oscillations predicts visual perception. *Journal of neuroscience*, *29*(24), 7869-7876.

---

### Decision · Program_Chairs · 2026-04-30

**Decision:**

Accept (regular)

**Comment:**

This work builds upon the Torus Graph (TG) model, an exponential-family distribution defined on phase variables that can represent conditional dependencies among brain oscillations. The authors develop a new a stochastic score matching approach and they prove convincingly that the new approach reduces the computational cost to enabling scalable inference for large models. The authors validated the new approach on real data, using brain experiments with Local Field recordings to demonstrate that the new approach can model high-dimensional phase interactions across brain states. Concerns and a number of weaknesses were openly acknowledged and resolved in the discussion phase.